# Uniform Concentration Bounds toward a Unified Framework for Robust Clustering

**Debolina Paul**[*]
Department of Statistics
Stanford University
debolinap@stanford.edu

**Saptarshi Chakraborty**[*]
Department of Statistics
University of California, Berkeley
saptarshic@berkeley.edu

**Swagatam Das**
Electronics & Communications Sciences Unit
Indian Statistical Institute
swagatam.das@isical.ac.in

**Jason Xu**[‡]
Department of Statistical Science
Duke University
jason.q.xu@duke.edu

## Abstract

Recent advances in center-based clustering continue to improve upon the drawbacks of Lloyd's celebrated $k$-means algorithm over 60 years after its introduction. Various methods seek to address poor local minima, sensitivity to outliers, and data that are not well-suited to Euclidean measures of fit, but many are supported largely empirically. Moreover, combining such approaches in a piecemeal manner can result in ad hoc methods, and the limited theoretical results supporting each individual contribution may no longer hold. Toward addressing these issues in a principled way, this paper proposes a cohesive robust framework for center-based clustering under a general class of dissimilarity measures. In particular, we present a rigorous theoretical treatment within a Median-of-Means (MoM) estimation framework, showing that it subsumes several popular $k$-means variants. In addition to unifying existing methods, we derive uniform concentration bounds that complete their analyses, and bridge these results to the MoM framework via Dudley's chaining arguments. Importantly, we neither require any assumptions on the distribution of the outlying observations nor on the relative number of observations $n$ to features $p$. We establish strong consistency and an error rate of $O(n^{-1/2})$ under mild conditions, surpassing the best-known results in the literature. The methods are empirically validated thoroughly on real and synthetic datasets.

## 1 Introduction

Clustering is a fundamental task in unsupervised learning, which seeks to discover naturally occurring groups within a dataset. Among a plethora of algorithms in the literature, center-based methods remain widely popular, with $k$-means [35, 33] as the most prominent example. Given $n$ data points $\mathcal{X} = \{\boldsymbol{X}_i : i = 1, \ldots, n\} \subset \mathbb{R}^p$, $k$-means seeks to partition the data into $k$ mutually exclusive and exhaustive groups by minimizing the within cluster variance. Representing the cluster centroids $\boldsymbol{\Theta} = \{\boldsymbol{\theta}_1, \ldots, \boldsymbol{\theta}_k\} \subset \mathbb{R}^p$, the $k$-means problem is formulated as the minimization of the objective function $f_{k\text{-means}}(\boldsymbol{\Theta}) = \sum_{i=1}^{n} \min_{1 \leq j \leq k} d(\boldsymbol{X}_i, \boldsymbol{\theta}_j)$, where $d(\cdot, \cdot)$ is a dissimilarity measure. Taking the squared Euclidean distance $d(\boldsymbol{x}, \boldsymbol{y}) = \|\boldsymbol{x} - \boldsymbol{y}\|_2^2$ yields the classical $k$-means formulation.

Clustering via $k$-means is well-documented to be sensitive to initialization [54], prone to getting stuck in poor local optima [57, 56, 17], and fragile against linearly non-separable clusters [39] or

---

[*]Joint first authors contributed equally     [‡] Corresponding author

35th Conference on Neural Information Processing Systems (NeurIPS 2021).

in the presence of even a single outlying observation [26]. Researchers continue to tackle these drawbacks of $k$-means while preserving its simplicity and interpretability [4, 1, 48, 40, 58, 49]. Although often successful in practice, few of these approaches are grounded in rigorous theory or provide finite-sample statistical guarantees. The available consistency results are mostly asymptotic in nature [17, 51, 52], lacking convergence rates or operating under restrictive assumptions on the relation between $p$ and $n$ [42]. For example, the recent large sample analysis [42] of the hard Bregman $k$-means algorithm [4] obtains an asymptotic error rate of $O(\sqrt{\log n/n})$ under restrictive assumptions on the relation between $n$ and $p$. The approach by [48] focuses on a unified framework from an optimization perspective, while [3, 49] focus on different divergence-based methods to better understand the underlying feature-space.

The presence of outliers only further complicates matters. Outlying data are common in real applications, but many of the aforementioned approaches are fragile to deviations from the assumed data generating mechanism. On the other hand, recent work on robust clustering methods [18, 21, 26] do not integrate the practical advances surveyed above, and similarly lack finite-sample guarantees. To bridge this gap, the Median of Means (MoM) literature provides a promising and attractive framework to robustify center-based clustering methods against outliers. MoM estimators are not only insensitive to outliers, but are also equipped with exponential concentration results under the mild condition of finite variance [34, 29, 6, 31, 27]. Recently, near-optimal results for mean estimation [37], classification [30], regression [36, 34], clustering [26, 13], bandits [14] and optimal transport [46] have been established from this perspective.

Under the MoM lens, we propose a unified framework for robust center-based clustering. Our treatment considers a family of Bregman loss functions not restricted to the classical squared Euclidean loss. We explore the exact sample error bounds for a general class of algorithms under mild regularity assumptions that apply to popular existing approaches, which we show to be special cases. The proposed framework allows for the data to be divided into two categories: the set of inliers ($\mathcal{I}$) and the set of outliers ($\mathcal{O}$). The inliers are assumed to be independent and identically distributed (i.i.d.), while absolutely no assumption is required on $\mathcal{O}$, allowing outliers to be unboundedly large, dependent on each other, sampled from a heavy-tailed distribution and so on. Our contributions within the MoM framework make use of Rademacher complexities [5, 7] and symmetrization arguments [53], powerful tools that often find use in the empirical process literature but, in our view, are underexplored in the context of robust clustering.

The paper is organized as follows. In Section 2, we identify a general centroid-based clustering framework which encompasses $k$-means [35], $k$-harmonic means [57], and power $k$-means [56] as special cases, to name a few. We show how this framework is made robust via Median of Means estimation, yielding an array of center-based robust clustering methods. Within this framework, we derive uniform deviation bounds and concentration inequalities under standard regularity conditions through bounding Rademacher complexity by metric entropy via Dudley's chaining argument in Section 3. The analysis newly reveals the convergence rate for popularly used clustering methods such as $k$-harmonic means and power $k$-means, matching the known rate results for $k$-means, and elegantly carries over to their MoM counterparts. We then implement and empirically assess the resulting algorithms through simulated and real data experiments. In particular, we find that power $k$-means [56] under the MoM paradigm outperforms the state-of-the-art in the presence of outliers.

## 1.1 Related theoretical analyses of clustering

In seminal work, Pollard [43], proved the strong consistency of $k$-means under a finite second moment assumption, spurring the large sample analysis of unsupervised clustering methods. This result has been extended to separable Hilbert spaces [10, 32] and for related algorithms [15, 51, 52], but these do not provide guarantees on the number of samples required so that the excess risk falls below a given threshold. Towards finding probabilistic error bounds, following research derived uniform concentration results for $k$-means and its variants [49], sub-Gaussian distortion bounds for the $k$-medians problem [12], and a $O(\sqrt{\log n/n})$ bound on $k$-means with Bregman divergences [41]. More recently, concentration inequalities for $k$-means under the MoM paradigm have been established [26, 13] under the restriction that sample cluster centroids ($\widehat{\Theta}_n$ in Section 3) are assumed to be bounded. This paper shows how a number of center-based clustering methods can be brought under the same umbrella and can be robustified using a general-purpose scheme. The theoretical analyses of this broad spectrum of methods is conducted via Dudley's chaining arguments and through the aid

of Rademacher complexity-based uniform concentration bounds. This approach enables us to replace assumptions on the sample cluster centroids by a bounded support assumption of the (inlying) data points, which yields an intuitive way of ensuring the boundedness of the cluster centroids through the obtuse angle property of Bregman divergences (Lemma 3.1 below). In contrast to the prior results, our analyses are *not* asymptotic in nature, so that the derived bounds hold for all values of the model parameters.

## 2   Problem Setting and Proposed Method

We consider the problem of partitioning a set of $n$ data points $\mathcal{X} = \{\boldsymbol{X}_i : i = 1, \ldots, n\} \subset \mathbb{R}^p$ into $k$ mutually exclusive clusters. In a center-based clustering framework, we represent the $j^{\text{th}}$ cluster by its centroid $\boldsymbol{\theta}_j \in \mathbb{R}^p$ for each $j \in \{1, \ldots, k\}$. To quantify the notion of "closeness", we allow the dissimilarity measure to be any Bregman divergence. Recall any differentiable, convex function $\phi : \mathbb{R}^p \to \mathbb{R}$ generates the Bregman divergence $d_\phi : \mathbb{R}^p \times \mathbb{R}^p \to \mathbb{R}_{\geq 0}$ ($\mathbb{R}_{\geq 0}$ denoting the set of non-negative reals) defined as

$$d_\phi(\boldsymbol{x}, \boldsymbol{y}) = \phi(\boldsymbol{x}) - \phi(\boldsymbol{y}) - \langle \nabla \phi(\boldsymbol{y}), \boldsymbol{x} - \boldsymbol{y} \rangle.$$

For instance, $\phi(\boldsymbol{u}) = \|\boldsymbol{u}\|_2^2$ generates the Euclidean distance. Without loss of generality, one may assume $\phi(\boldsymbol{0}) = \nabla \phi(\boldsymbol{0}) = 0$. In this paradigm, clustering is achieved by minimizing the objective

$$\frac{1}{n} \sum_{i=1}^{n} \Psi_{\boldsymbol{\alpha}} \left( d_\phi(\boldsymbol{X}_i, \boldsymbol{\theta}_1), \ldots, d_\phi(\boldsymbol{X}_i, \boldsymbol{\theta}_k) \right) := f_{\boldsymbol{\Theta}}(\boldsymbol{X}). \tag{1}$$

Here $\Psi_{\boldsymbol{\alpha}} : \mathbb{R}_{\geq 0}^k \to \mathbb{R}_{\geq 0}$ is a component-wise non-decreasing function (such as a generalized mean) of the dissimilarities $\{d_\phi(\boldsymbol{X}, \boldsymbol{\Theta}_j)\}_{j=1}^k$ which satisfies $\Psi(\boldsymbol{0}) = 0$. The hyperparameter $\boldsymbol{\alpha} \in \mathcal{A} \subseteq \mathbb{R}^q$ is specified by the user, and we will additionally assume that $\Psi$ is Lipschitz. For intuition, we begin by showing how this setup includes several popular clustering methods.

**Examples:**   Suppose $\phi(\boldsymbol{u}) = \|\boldsymbol{u}\|_2^2$ and $\Psi(\boldsymbol{x}) = (\sum_{j=1}^k x_j^{-1})^{-1}$. Then the objective (1) becomes $\frac{1}{n} \sum_{i=1}^n (\sum_{j=1}^k \|\boldsymbol{X}_i - \boldsymbol{\theta}_j\|_2^{-2})^{-1}$, which is the objective function of $k$-harmonic means clustering [57]. Now consider other generalized means: take $\Psi_s(\boldsymbol{x}) = M_s(\boldsymbol{x})$ where we denote the *power mean* $M_s(\boldsymbol{x}) = (k^{-1} \sum_{i=1}^k x_i^s)^{1/s}$ for $s \in (-\infty, -1]$. Then objective (1) coincides with the recent power $k$-means method [56], $\frac{1}{n} \sum_{i=1}^n M_s \left( \|\boldsymbol{X}_i - \boldsymbol{\theta}_1\|_2^2, \ldots, \|\boldsymbol{X}_i - \boldsymbol{\theta}_k\|_2^2 \right)$. When $\Psi(\boldsymbol{x}) = \min_{1 \leq j \leq k} x_j$, (1) recovers Bregman hard clustering $\frac{1}{n} \sum_{i=1}^n \min_{1 \leq j \leq k} d_\phi(\boldsymbol{X}_i, \boldsymbol{\theta}_j)$ proposed in [4] for any valid $\phi$, while the special case of $\phi(\boldsymbol{u}) = \|\boldsymbol{u}\|_2^2$ yields the familiar Euclidean $k$-means problem [35].

In what follows, we derive concentration bounds that establish new theoretical guarantees such as consistency and convergence rates for clustering algorithms in this framework. These analyses lead us to a unified, robust framework by embedding this class of methods within the Median of Means (MoM) estimation paradigm. Via elegant connections between the properties of MoM estimators and Vapnik-Chervonenkis (VC) theory, our MoM estimators too will inherit uniform concentration inequalities from the preceding analysis, extending convergence guarantees to the robust setting.

**Median of Means**   Instead of directly minimizing the empirical risk (1), MoM begins by partitioning the data into $L$ sets $B_1, \ldots, B_L$ which each contain exactly $b$ many elements (discarding a few observations when $n$ is not divisible by $L$). The partitions can be assigned uniformly at random, or can be shuffled throughout the algorithm [30]. MoM then entails a robust version of the estimator defined under (1) by instead minimizing the following objective with respect to $\boldsymbol{\Theta}$:

$$\text{MoM}_L^n(\boldsymbol{\Theta}) = \text{Median} \left( \frac{1}{b} \sum_{i \in B_1} f_{\boldsymbol{\Theta}}(\boldsymbol{X}_i), \ldots, \frac{1}{b} \sum_{i \in B_L} f_{\boldsymbol{\Theta}}(\boldsymbol{X}_i) \right). \tag{2}$$

Intuitively, MoM estimators are more robust than their ERM counterparts since under mild conditions, only a subset of the partitions is contaminated by outliers while others are outlier-free. Taking the median over partitions negates the influence of these spurious partitions, thus reducing the effect of outliers. Formal analysis of robustness via breakdown points is also available for MoM estimators [28, 44]; we will make use of the nice concentration properties of MoM estimators in Section 3.4.

**Algorithm 1** MoM Clustering via Adagrad

---

**Input:** $\boldsymbol{X} \in \mathbb{R}^{n \times p}$, $k$, $L$, $f_{\boldsymbol{\Theta}}(\cdot)$, $\eta$, $\epsilon$.
**Output:** The cluster centroids $\boldsymbol{\Theta}$.
Initialization: Randomly partition $\{1, \ldots, n\}$ into $L$ many partitions of equal length. Randomly choose $k$ points without replacement from $\mathcal{X}$ to initialize $\boldsymbol{\Theta}_0$.
**repeat**
    **Step 1:** Find $\ell_t \in \{1, \ldots, L\}$, such that $\mathrm{MoM}_L^n(\boldsymbol{\Theta}_t) = \frac{1}{b} \sum_{i \in B_{\ell_t}} f_{\boldsymbol{\Theta}_t}(\boldsymbol{X}_i)$.

    **Step 2:** $\boldsymbol{g}_j^{(t)} \leftarrow \frac{1}{b} \sum_{i \in B_{\ell_t}} \nabla_{\boldsymbol{\theta}_j} f_{\boldsymbol{\Theta}}(\boldsymbol{X}_i)$

    **Step 3:** Update $\boldsymbol{\Theta}$ by $\boldsymbol{\theta}_j^{(t+1)} \leftarrow \boldsymbol{\theta}_j^{(t)} - \frac{\eta}{\sqrt{\epsilon + \sum_{t'=1}^{(t)} \|\boldsymbol{g}_j^{(t')}\|_2^2}} \boldsymbol{g}_j^{(t)}$.

**until** objective (2) converges

---

Furthermore, optimizing (2) is made tractable via gradient-based methods. We advocate the Adagrad algorithm [19, 22], whose updates are given by

$$\boldsymbol{\theta}_j^{(t+1)} \leftarrow \boldsymbol{\theta}_j^{(t)} - \frac{\eta}{\sqrt{\epsilon + \sum_{t'=1}^{(t)} \|\boldsymbol{g}_j^{(t')}\|_2^2}} \boldsymbol{g}_j^{(t)}$$

for hyperparameter $\epsilon > 0$, learning rate $\eta > 0$, and $\boldsymbol{g}_j^{(t)}$ denoting a subgradient of $\mathrm{MoM}_L^n(\boldsymbol{\Theta})$ at $\boldsymbol{\Theta}_t$. That is, if $\ell_t$ denotes the median partition at step $t$, then

$$\nabla_{\boldsymbol{\Theta}_j} \mathrm{MoM}_L^n(\boldsymbol{\Theta}_t) = \frac{1}{b} \sum_{i \in B_{\ell_t}} \nabla_{\boldsymbol{\theta}_j} f_{\boldsymbol{\Theta}}(\boldsymbol{X}_i)|_{\boldsymbol{\Theta} = \boldsymbol{\Theta}_t}.$$

For any $\Psi_{\boldsymbol{\alpha}}$ differentiable, $\nabla_{\boldsymbol{\theta}_j} f_{\boldsymbol{\Theta}}(\boldsymbol{x}) = \partial_j \Psi_{\boldsymbol{\alpha}}\left(d_{\phi}(\boldsymbol{x}, \boldsymbol{\theta}_1^{(t)}), \ldots, d_{\phi}(\boldsymbol{x}, \boldsymbol{\theta}_k^{(t)})\right) \nabla_{\boldsymbol{\theta}} d_{\phi}(\boldsymbol{x}, \boldsymbol{\theta})$ by the chain rule. As a concrete illustration of the general formula, consider the power $k$-means objective $f_{\boldsymbol{\Theta}}(\boldsymbol{x}) = M_s(\|\boldsymbol{x} - \boldsymbol{\theta}_1\|_2^2, \ldots, \|\boldsymbol{x} - \boldsymbol{\theta}_k\|_2^2)$: upon partial differentiation, we obtain

$$\nabla_{\boldsymbol{\theta}_j} f_{\boldsymbol{\Theta}}(\boldsymbol{x}) = \frac{2}{k} \left(\frac{1}{k} \sum_{j'=1}^{k} \|\boldsymbol{x} - \boldsymbol{\theta}_{j'}\|_2^{2s}\right)^{1/s-1} \|\boldsymbol{x} - \boldsymbol{\theta}_j\|_2^{2(s-1)} (\boldsymbol{\theta}_j - \boldsymbol{x}).$$

As another example, the classic $k$-means objective $f_{\boldsymbol{\Theta}}(\boldsymbol{x}) = \min_{\boldsymbol{\theta} \in \boldsymbol{\Theta}} \|\boldsymbol{x} - \boldsymbol{\theta}\|_2^2$ requires the subgradient $\nabla_{\boldsymbol{\theta}_j} f_{\boldsymbol{\Theta}}(\boldsymbol{x}) = 2(\boldsymbol{\theta}_j - \boldsymbol{x}) \mathbb{1}\{j \in \mathcal{J}(\boldsymbol{\Theta}, \boldsymbol{x})\}$, where $\mathcal{J}(\boldsymbol{\Theta}, \boldsymbol{x}) = \mathrm{argmin}_{1 \leq j \leq k} \|\boldsymbol{x} - \boldsymbol{\theta}_j\|_2^2$ and $\mathbb{1}\{\cdot\}$ denotes the indicator function.

A pseudocode summary appears in Algorithm 1. This method tends to find clusters efficiently: for instance, we incur $O(npk)$ per-iteration complexity applied to both the $k$-means and power $k$-means instances, which matches that of their original algorithms without robustifying via MoM [35, 33, 56]. Of course, here we update $\boldsymbol{\theta}_j$ by an adaptive gradient step rather than by a closed form expression.

As emphasized by an anonymous reviewer, one should note that the algorithms under the proposed framework may converge to sub-optimal local solutions under the proposed optimization scheme. As is the case for $k$-means and its variants, this possibility arises due to the non-convexity of the objective functions (1) and (2). A complete theoretical understanding from an optimization perspective for such methods is not yet fully developed and global results are in general notoriously difficult to obtain. Having said that, it is worth noting that recent empirical analysis show that techniques such as annealing [56, 17] may be effective to circumvent this difficulty. Indeed, our experimental analysis (see Section 4) suggests that a robust version of power $k$-means using this annealing technique best overcomes this difficulty even in the presence of outliers.

## 3 Theoretical Analysis

Here we analyze properties of the proposed framework (1), with complete details of all proofs in the Supplement. Denoting $\mathcal{M}$ the set of all probability measures $P$ with support on $[-M, M]^p$, i.e. $P([-M, M]^p) = 1$, we first make standard assumptions that the data are i.i.d. with bounded components [9, 17, 42].

**A1.** $\boldsymbol{X}_1 \ldots, \boldsymbol{X}_n \overset{i.i.d.}{\sim} P$ *such that* $P \in \mathcal{M}$.

Let $P_n$ be the empirical distribution based of the data $\boldsymbol{X}_1 \ldots, \boldsymbol{X}_n$. That is, for any Borel set $A$, $P_n(A) = \frac{1}{n} \sum_{i=1}^{n} \mathbb{1}\{\boldsymbol{X}_i \in A\}$. For notational simplicity, we write $\mu f := \int f d\mu$. Appealing to the Strong Law of Large Numbers (SLLN) [2], $P_n f_{\boldsymbol{\Theta}} \xrightarrow{a.s.} P f_{\boldsymbol{\Theta}}$. Suppose $\widehat{\boldsymbol{\Theta}}_n$ be a (global) minimizer of (1) and $\boldsymbol{\Theta}^*$ be the global minimizer of $P f_{\boldsymbol{\Theta}}$. Since the functions $P_n f_{\boldsymbol{\Theta}}$ and $P f_{\boldsymbol{\Theta}}$ are close to each other as $n$ become large, we can expect that their respective minimizers, $\widehat{\boldsymbol{\Theta}}_n$ and $\boldsymbol{\Theta}^*$ are also close to one another. To show that $\widehat{\boldsymbol{\Theta}}_n$ converges to $\boldsymbol{\Theta}^*$ as $n \to \infty$, we consider bounding the uniform deviation $\sup_{\boldsymbol{\Theta}} |P_n f_{\boldsymbol{\Theta}} - P f_{\boldsymbol{\Theta}}|$. Towards establishing such bounds, we will posit two regularity assumptions on $\Psi_{\boldsymbol{\alpha}}(\cdot)$ and $\phi(\cdot)$, beginning with a $\tau_{\boldsymbol{\alpha},k}$-Lipschitz condition on $\Psi_{\boldsymbol{\alpha}}$.

**A2.** *For all* $\boldsymbol{\alpha} \in \mathcal{A}$ *and any* $\boldsymbol{x}, \boldsymbol{y} \in \mathbb{R}^k_{\geq 0}$, *we have* $|\Psi_{\boldsymbol{\alpha}}(\boldsymbol{x}) - \Psi_{\boldsymbol{\alpha}}(\boldsymbol{y})| \leq \tau_{\boldsymbol{\alpha},k} \|\boldsymbol{x} - \boldsymbol{y}\|_1$.

We also assume a weaker form of a standard condition that the gradient $\nabla \phi(\cdot)$ is $H_p$-Lipschitz [50]; unlike their work, note we do *not* additionally require strong convexity of $\phi$.

**A3.** *There exists* $H_p \geq 0$ *such that* $\|\nabla \phi(\boldsymbol{x}) - \nabla \phi(\boldsymbol{y})\|_2 \leq H_p \|\boldsymbol{x} - \boldsymbol{y}\|_2$ *for all* $\boldsymbol{x}, \boldsymbol{y} \in [-M, M]^p$.

These conditions are mild, and can be seen to hold for all of the aforementioned popular special cases. For instance, taking $\phi(\boldsymbol{u}) = \|\boldsymbol{u}\|^2$ yields the squared Euclidean distances, and we see that $\nabla \phi(\boldsymbol{u}) = 2\boldsymbol{u}$ so that A3 is satisfied with constant $H_p = 2$. Now, let $\Psi(\boldsymbol{x}) = \min_{1 \leq j \leq k} x_j$ as in classical $k$-means, and denote $j^* \in \arg\min_{1 \leq j \leq k} y_j$. For any vectors $\boldsymbol{x}, \boldsymbol{y} \in \mathbb{R}^p_{\geq 0}$,

$$\Psi(\boldsymbol{x}) - \Psi(\boldsymbol{y}) = \min_{1 \leq j \leq k} x_j - \min_{1 \leq j \leq k} y_j = \min_{1 \leq j \leq k} x_j - y_{j^*} \leq x_{j^*} - y_{j^*} \leq \|\boldsymbol{x} - \boldsymbol{y}\|_1.$$

Thus, $\Psi$ is clearly non-negative and componentwise non-decreasing, and satisfies A2. Similarly, if we take $\Psi_s(\boldsymbol{x}) = M_s(\boldsymbol{x})$, the conditions are met for power $k$-means: it is again non-negative and non-decreasing in its components, and satisfies A2 with $\tau_{\boldsymbol{\alpha},k} = k^{-1/s}$ due to [8].

### 3.1 Bounds on $\widehat{\boldsymbol{\Theta}}_n$ and $\boldsymbol{\Theta}^*$

Toward proving that $\widehat{\boldsymbol{\Theta}}_n$ converges to $\boldsymbol{\Theta}^*$, we will need to show that both $\widehat{\boldsymbol{\Theta}}_n$ and $\boldsymbol{\Theta}^*$ lie in $[-M, M]^{k \times p}$. To this end, we first establish the obtuse angle property for Bregman divergences.

**Lemma 3.1.** *Let $\mathcal{C}$ be a convex set and suppose $P_{\mathcal{C}}(\boldsymbol{\theta})$ be the projection of $\boldsymbol{\theta}$ onto $\mathcal{C}$, with respect to the Bregman divergence $d_\phi(\cdot, \cdot)$, i.e. $P_{\mathcal{C}}(\boldsymbol{\theta}) = \arg\min_{\boldsymbol{x} \in \mathcal{C}} d_\phi(\boldsymbol{x}, \boldsymbol{\theta})$ (assuming it exists). Then,*

$$d_\phi(\boldsymbol{x}, \boldsymbol{\theta}) \geq d_\phi(\boldsymbol{x}, P_{\mathcal{C}}(\boldsymbol{\theta})) + d_\phi(P_{\mathcal{C}}(\boldsymbol{\theta}), \boldsymbol{\theta}), \quad \text{for all } \boldsymbol{x} \in \mathcal{C}.$$

We next show that to minimize $P_n f_{\boldsymbol{\Theta}}$ or $P f_{\boldsymbol{\Theta}}$, it is enough to restrict the search to $[-M, M]^{k \times p}$.

**Lemma 3.2.** *Let A2 hold, and $Q \in \mathcal{M}$. Let $d_\phi : \mathbb{R}^p \times \mathbb{R}^p \to \mathbb{R}_{\geq 0}$ be a Bregman divergence. Then for any $\boldsymbol{\Theta} \in \mathbb{R}^{k \times p}$, there exists $\boldsymbol{\Theta}' \in [-M, M]^{k \times p}$, such that $Q f_{\boldsymbol{\Theta}'} \leq Q f_{\boldsymbol{\Theta}}$.*

Since we can restrict our attention to $[-M, M]^{k \times p}$ to minimize $Q f_{\boldsymbol{\Theta}}$, we have the following:

**Corollary 3.1.** *Let $Q \in \mathcal{M}$ and $d_\phi : \mathbb{R}^p \times \mathbb{R}^p \to \mathbb{R}_{\geq 0}$ be a Bregman divergence. If $\boldsymbol{\Theta}_0 = \arg\min_{\boldsymbol{\Theta} \in \mathbb{R}^{k \times p}} \int f_{\boldsymbol{\Theta}} dQ$, then $\boldsymbol{\Theta}_0 \in [-M, M]^{k \times p}$.*

Now note under A1 both $P$ and $P_n$ have support contained in $[-M, M]^{k \times p}$. The following corollary is thus implied by replacing $Q$ by $P$ and $P_n$.

**Corollary 3.2.** *Under A1—3, both $\widehat{\boldsymbol{\Theta}}_n, \boldsymbol{\Theta}^* \in [-M, M]^{k \times p}$.*

Now that we have bounded $\widehat{\boldsymbol{\Theta}}_n$ and $\boldsymbol{\Theta}^*$ in a compact set, the following section supplies probabilistic bounds on the uniform deviation $2 \sup_{\boldsymbol{\Theta} \in [-M,M]^{k \times p}} |P_n f_{\boldsymbol{\Theta}} - P f_{\boldsymbol{\Theta}}|$ via metric entropy arguments.

### 3.2 Concentration Inequality and Metric Entropy Bounds via Rademacher Complexity

We have proven that $\widehat{\boldsymbol{\Theta}}_n, \boldsymbol{\Theta}^* \in [-M, M]^{k \times p}$. To bound the difference $|P f_{\widehat{\boldsymbol{\Theta}}_n} - P f_{\boldsymbol{\Theta}^*}|$, we observe

$$\begin{aligned}
|P f_{\widehat{\boldsymbol{\Theta}}_n} - P f_{\boldsymbol{\Theta}^*}| &= P f_{\widehat{\boldsymbol{\Theta}}_n} - P f_{\boldsymbol{\Theta}^*} = P f_{\widehat{\boldsymbol{\Theta}}_n} - P_n f_{\widehat{\boldsymbol{\Theta}}_n} + P_n f_{\widehat{\boldsymbol{\Theta}}_n} - P_n f_{\boldsymbol{\Theta}^*} + P_n f_{\boldsymbol{\Theta}^*} - P f_{\boldsymbol{\Theta}^*} \\
&\leq P f_{\widehat{\boldsymbol{\Theta}}_n} - P_n f_{\widehat{\boldsymbol{\Theta}}_n} + P_n f_{\boldsymbol{\Theta}^*} - P f_{\boldsymbol{\Theta}^*} \leq 2 \sup_{\boldsymbol{\Theta} \in [-M,M]^{k \times p}} |P_n f_{\boldsymbol{\Theta}} - P f_{\boldsymbol{\Theta}}|. \quad (3)
\end{aligned}$$

Thus, to bound $|Pf_{\widehat{\Theta}_n} - Pf_{\Theta^*}|$, it is enough to prove bounds on $\sup_{\Theta \in [-M,M]^{k \times p}} |P_n f_{\Theta} - Pf_{\Theta}|$. The main idea here is to bound this uniform deviation via Rademacher complexity, and in turn bound the Rademacher complexity itself [20, 38]. Let $\mathcal{F} = \{f_{\Theta} : \Theta \in [-M,M]^{k \times p}\}$, and denote the $\mathcal{F}$-norm [2] between two probability measures $\mu$ and $\nu$ as $\|\mu - \nu\|_{\mathcal{F}} = \sup_{f \in \mathcal{F}} |\int f d\mu - \int f d\nu|$. We recall the definition of Rademacher complexity and covering number as follows.

**Definition 1.** *Let $\epsilon_i$'s be i.i.d. Rademacher random variables independent of $\mathcal{X} = \{X_1, \ldots, X_n\}$, i.e. $\mathbb{P}(\epsilon_i = 1) = \mathbb{P}(\epsilon_i = -1) = 0.5$, The population Rademacher complexity of $\mathcal{F}$ is defined as $\mathcal{R}_n(\mathcal{F}) = \mathbb{E} \sup_{f \in \mathcal{F}} \frac{1}{n} \sum_{i=1}^n \epsilon_i f(X_i)$, where the expectation is over both $\epsilon$ and $\mathcal{X}$.*

**Definition 2.** *($\delta$-cover and Covering Number) Let $(X, d)$ be a metric space. The set $X_{\delta} \subseteq X$ is said to be a $\delta$-cover of $X$ if for all $x \in X$, $\exists\, x' \in X_{\delta}$, such that $d(x, x') \leq \delta$. The $\delta$-covering number of $X$ w.r.t. $d$, denoted by $N(\delta; X, d)$, is the size of the smallest $\delta$-cover of $X$ with respect to d.*

The following Lemma gives a bound for the $\delta$-covering number of $\mathcal{F}$ under the supremum norm. The main idea here is to use the Lipschitz property of $f_{\Theta}$ and then to find a cover of the search space for $\Theta$, i.e. $[-M, M]^{k \times p}$. This then automatically transcends to a cover of $\mathcal{F}$ under the sup-norm.

**Lemma 3.3.** *Let $N(\delta; \mathcal{F}, \|\cdot\|_{\infty})$ be the $\delta$-covering number of $\mathcal{F}$ under $\|\cdot\|_{\infty}$. Then, under A1—3,*

$$N(\delta; \mathcal{F}, \|\cdot\|_{\infty}) \leq \left( \max \left\{ \left\lfloor \frac{8M^2 \tau_{\boldsymbol{\alpha},k} H_p kp}{\delta} \right\rfloor, 1 \right\} \right)^{kp}.$$

To bound the Rademacher complexity, we will also need to bound the diameter of $\mathcal{F}$ under $\|\cdot\|_{\infty}$:

**Lemma 3.4.** *Let $diam(\mathcal{F}) = \sup_{f,g \in \mathcal{F}} \|f - g\|_{\infty}$. Then, under A1—3, $diam(\mathcal{F}) \leq 8\tau_{\boldsymbol{\alpha},k} H_p M^2 kp$.*

We are now ready to state the bound on the Rademacher complexity $\mathcal{R}_n(\mathcal{F})$ in Theorem 3.1. We provide a sketch of the argument below, with the complete proof details available in the Supplement.

**Theorem 3.1.** *Under A1—3, $\mathcal{R}_n(\mathcal{F}) \leq 48\sqrt{\pi}\tau_{\boldsymbol{\alpha},k} H_p M^2 (kp)^{3/2} n^{-1/2}$.*

**Proof Sketch** The main idea of the proof is to use Dudley's chaining argument [20, 55] to bound the Rademacher complexity in terms of an integral involving the metric entropy $\log N(\delta; \mathcal{F}, \|\cdot\|_{\infty})$. Let $\Delta = 8H_p M^2 k^{1-1/s} p$. Using the chaining approach, one can show that

$$\mathcal{R}_n(\mathcal{F}) \leq \frac{12}{\sqrt{n}} \int_0^{\Delta} \sqrt{\log N(\epsilon; \mathcal{F}, \|\cdot\|_{\infty})} d\epsilon \leq \frac{12}{\sqrt{n}} \int_0^{\Delta} \sqrt{kp \log \left( \max \left\{ \frac{\Delta}{\epsilon}, 1 \right\} \right)} d\epsilon \quad (4)$$

$$= \frac{12}{\sqrt{n}} \int_0^{\Delta} \sqrt{kp \log \left( \frac{\Delta}{\epsilon} \right)} d\epsilon$$

$$= 48\sqrt{\pi}\tau_{\boldsymbol{\alpha},k} H_p M^2 (kp)^{3/2} n^{-1/2}.$$

Here the second inequality (4) follows from Lemma 3.3. Before we proceed, we state the following Lemma, which puts an uniform bound on $\|f\|_{\infty}$, $f \in \mathcal{F}$.

**Lemma 3.5.** *For all $\boldsymbol{x} \in [-M, M]^p$ and $\Theta \in [-M, M]^{k \times p}$,*

$$0 \leq \Psi_{\boldsymbol{\alpha}}(d_{\phi}(\boldsymbol{x}, \boldsymbol{\theta}_1), \ldots, d_{\phi}(\boldsymbol{x}, \boldsymbol{\theta}_k)) \leq 4\tau_{\boldsymbol{\alpha},k} H_p M^2 pk.$$

Now that we have established a bound on the Rademacher complexity $\mathcal{R}_n(\mathcal{F})$, we are now ready to establish a uniform concentration inequality on $\|P_n - P\|_{\mathcal{F}} \triangleq \sup_{f \in \mathcal{F}} |P_n f - Pf|$ in the following Theorem, proven in the Supplement.

**Theorem 3.2.** *Under A1—3, with probability at least $1 - \delta$, the following holds.*

$$\|P_n - P\|_{\mathcal{F}} \leq 96\sqrt{\pi}\tau_{\boldsymbol{\alpha},k} H_p M^2 (kp)^{3/2} n^{-1/2} + 4\tau_{\boldsymbol{\alpha},k} H_p M^2 pk \sqrt{\frac{\log(2/\delta)}{2n}}.$$

The result in Theorem 3.2, reveals a non-asymptotic bound on $|Pf_{\widehat{\Theta}_n} - Pf_{\Theta^*}|$:

**Corollary 3.3.** *Under A1—3, with probability at least $1 - \delta$, the following holds.*

$$|Pf_{\widehat{\Theta}_n} - Pf_{\Theta^*}| \leq 192\sqrt{\pi}\tau_{\boldsymbol{\alpha},k} H_p M^2 (kp)^{3/2} n^{-1/2} + 8\tau_{\boldsymbol{\alpha},k} H_p M^2 pk \sqrt{\frac{\log(2/\delta)}{2n}}.$$

*Proof.* From equation (3), we know that $|Pf_{\widehat{\Theta}_n} - Pf_{\Theta^*}| \leq 2\|P_n - P\|_{\mathcal{F}}$. The corollary follows immediately by application of Theorem 3.2. $\qquad\square$

## 3.3 Inference for Fixed $p$: Strong and $\sqrt{n}$ Consistency

We now discuss the classical domain where $p$ is kept fixed [43, 17, 51] and show that the results above imply strong and $\sqrt{n}$-consistency, mirroring some known results for existing methods such as $k$-means. We first solidify the notion of convergence of the centroids $\widehat{\boldsymbol{\Theta}}_n$ to $\boldsymbol{\Theta}^*$, following Pollard [43]. Since centroids are unique only up to label permutations, our notion of dissimilarity

$$\text{diss}(\boldsymbol{\Theta}_1, \boldsymbol{\Theta}_2) = \min_{M \in \mathcal{P}_k} \|\boldsymbol{\Theta}_1 - M\boldsymbol{\Theta}_2\|_F$$

is considered over $\mathcal{P}_k$ the set of all $k \times k$ real permutation matrices, where $\|\cdot\|_F$ denotes the Frobenius norm. Now, we say that the sequence $\boldsymbol{\Theta}_n$ converges to $\boldsymbol{\Theta}$ if $\lim_{n \to \infty} \text{diss}(\boldsymbol{\Theta}_n, \boldsymbol{\Theta}) = 0$. We begin by imposing the following standard identifiablity condition [43, 51, 17] on $P$ for our analysis.

**A4.** *For all $\eta > 0$, there exists $\epsilon > 0$, such that $Pf_{\boldsymbol{\Theta}} > Pf_{\boldsymbol{\Theta}^*} + \epsilon$ whenever $\text{diss}(\boldsymbol{\Theta}, \boldsymbol{\Theta}^*) > \eta$.*

We now investigate the strong consistency properties of $\widehat{\boldsymbol{\Theta}}_n$, and also investigate the rate at which $|Pf_{\widehat{\boldsymbol{\Theta}}_n} - Pf_{\boldsymbol{\Theta}^*}|$ converges to 0. Theorem 3.3 states that indeed strong consistency holds, with convergence rate $O(n^{-1/2})$. Note that this rate is faster than that found previously in [42]. Before we proceed, recall we say that $X_n = O_P(a_n)$ if the sequence $X_n/a_n$ is tight [2].

**Theorem 3.3.** *(Strong consistency and $\sqrt{n}$-consistency) If $p$ is kept fixed then under A1—4, $\widehat{\boldsymbol{\Theta}}_n \xrightarrow{a.s.} \boldsymbol{\Theta}^*$ under $P$. Moreover, $|Pf_{\widehat{\boldsymbol{\Theta}}_n} - Pf_{\boldsymbol{\Theta}^*}| = O_P(n^{-1/2})$.*

**Note:** Here $A_n$ is $O_P(a_n)$ means that $\{A_n/a_n\}_{n \in \mathbb{N}}$ is stochastically bounded or *tight*.

## 3.4 Inference Under the MoM Framework

Theorem 3.3 and the bounds we present above already establish novel statistical results that pertain to methods under (1) such as power $k$-means in the "uncontaminated" setting. In this section, we now extend these findings to the MoM setup under (2) in order to understand their behavior in the presence of outliers. Recall that in this setting, the data are partitioned into $L$ equally sized blocks; without loss of generality, we take $n = L \cdot b$. We denote the set of all inliers by $\{\boldsymbol{X}_i\}_{i \in \mathcal{I}}$ and outliers by $\{\boldsymbol{X}_i\}_{i \in \mathcal{O}}$. Now let $\widehat{\boldsymbol{\Theta}}_n^{(\text{MoM})}$ denote the minimizer of (2): towards establishing the error rate at which $|Pf_{\widehat{\boldsymbol{\Theta}}_n^{(\text{MoM})}} - Pf_{\boldsymbol{\Theta}^*}|$ goes to 0, we assume the following:

**A5.** $\{\boldsymbol{X}_i\}_{i \in \mathcal{I}} \overset{i.i.d.}{\sim} P$ with $P \in \mathcal{M}$.
**A6.** *There exists $\eta > 0$ such that $L > (2 + \eta)|\mathcal{O}|$.*

We remark A5 is identical to A1, but imposed only on the inlying observations. A6 ensures at least half of the $L$ partitions are free of outliers; note this is much weaker than requiring $L > 4|\mathcal{O}|$ as is done in recent work [30]. Importantly, we emphasize that *no distributional assumptions* regarding the outlying observations are made, allowing them to be unbounded, generated from heavy-tailed distributions, or dependent among each other. Proofs of the following results appear in the Supplement.

As a point of departure, we first establish that $\widehat{\boldsymbol{\Theta}}_n^{(\text{MoM})} \in [-M, M]^{k \times p}$.

**Lemma 3.6.** *Let A2—3 and A5—6 hold. Then for any $\boldsymbol{\Theta} \in \mathbb{R}^{k \times p}$, there exists $\boldsymbol{\Theta}' \in [-M, M]^{k \times p}$, such that $MoM_L^n(\boldsymbol{\Theta}') \leq MoM_L^n(\boldsymbol{\Theta})$.*

Again, we may restrict the search space for finding $\widehat{\boldsymbol{\Theta}}_n^{(\text{MoM})}$ in $[-M, M]^{k \times p}$ due to Lemma 3.6.

**Corollary 3.4.** *Under A2—3 and A5—6, $\widehat{\boldsymbol{\Theta}}_n^{(MoM)} \in [-M, M]^{k \times p}$.*

Similarly to Section 3, we derive a uniform bound on $\sup_{\boldsymbol{\Theta} \in [-M,M]^{k \times p}} |MoM_L^n(f_{\boldsymbol{\Theta}}) - Pf_{\boldsymbol{\Theta}}|$ and then bound $|Pf_{\widehat{\boldsymbol{\Theta}}_n^{(\text{MoM})}} - Pf_{\boldsymbol{\Theta}^*}|$ in turn. For brevity we define $\delta := 2/(4 + \eta) - |\mathcal{O}|/L$, and use the notation "$\lesssim$" to suppress the absolute constants. The uniform deviation bound is as follows, with complete proof appearing in the Supplement.

**Theorem 3.4.** *Under A2—3 and A5—6, with probability at least $1 - 2e^{-2L\delta^2}$, the following holds.*

$$\sup_{\boldsymbol{\Theta} \in [-M,M]^{k \times p}} |MoM_L^n(f_{\boldsymbol{\Theta}}) - Pf_{\boldsymbol{\Theta}}| \lesssim \tau_{\boldsymbol{\alpha},k} H_p \max \left\{ kp\sqrt{\frac{L}{n}}, (kp)^{3/2} \frac{\sqrt{|\mathcal{I}|}}{n} \right\}.$$

We give a brief outline of our proof for Theorem 3.4 below, with details appearing in the Supplement.

**Proof sketch**   We begin by noting that if $\sup_{\boldsymbol{\Theta} \in [-M,M]^{k \times p}} \sum_{\ell=1}^{L} \mathbb{1}\{(P - P_{B_\ell})f_{\boldsymbol{\Theta}} > \epsilon\} > \frac{L}{2}$, then $\sup_{\boldsymbol{\Theta} \in [-M,M]^{k \times p}}(Pf_{\boldsymbol{\Theta}} - \mathrm{MoM}_L^n(f_{\boldsymbol{\Theta}})) > \epsilon$ , where $P_{B_\ell}$ denotes the empirical distribution of $\{\boldsymbol{X}_i\}_{i \in B_\ell}$. This implies it is suffices to bound the quantity

$$\mathbb{P}\left(\sup_{\boldsymbol{\Theta} \in [-M,M]^{k \times p}} \sum_{\ell=1}^{L} \mathbb{1}\{(P - P_{B_\ell})f_{\boldsymbol{\Theta}} > \epsilon\} > \frac{L}{2}\right).$$

Introducing the function $\varphi(t) = (t-1)\mathbb{1}\{1 \leq t \leq 2\} + \mathbb{1}\{t > 2\}$, we begin by bounding outlier-free partitions, making use of the inequalities $\mathbb{1}\{t \geq 2\} \leq \varphi(t) \leq \mathbb{1}\{t \geq 1\}$. We then proceed by bounding $\sup_{\boldsymbol{\Theta} \in [-M,M]^{k \times p}} \sum_{\ell=1}^{L} \mathbb{1}\{(P - P_{B_\ell})f_{\boldsymbol{\Theta}} > \epsilon\}$ by the sum $\xi_1 + \xi_2 + |\mathcal{O}|$, where $\xi_1 = \sup_{\boldsymbol{\Theta} \in [-M,M]^{k \times p}} \sum_{\ell \in \mathcal{L}} \mathbb{E}\varphi\left(\frac{2(P - P_{B_\ell})f_{\boldsymbol{\Theta}}}{\epsilon}\right)$ and $\xi_2 = \sup_{\boldsymbol{\Theta} \in [-M,M]^{k \times p}} \sum_{\ell \in \mathcal{L}} \left[\varphi\left(\frac{2(P - P_{B_\ell})f_{\boldsymbol{\Theta}}}{\epsilon}\right) - \mathbb{E}\varphi\left(\frac{2(P - P_{B_\ell})f_{\boldsymbol{\Theta}}}{\epsilon}\right)\right]$. We bound $\xi_1$ by appealing to Hoeffding's inequality, while we show that $\xi_2$ can be bounded by applying the bounded difference inequality together with the result of Theorem 3.1 to bound the resulting Rademacher complexity.

The corollary below follows from this uniform bound, giving a non-asymptotic control over the difference $|Pf_{\widehat{\boldsymbol{\Theta}}_n^{(\text{MoM})}} - Pf_{\boldsymbol{\Theta}^*}|$ in terms of the model parameters.

**Corollary 3.5.** *Under A2—3 and A5—6, with probability at least $1 - 2e^{-2L\delta^2}$, the following holds.*

$$|Pf_{\widehat{\boldsymbol{\Theta}}_n^{(MoM)}} - Pf_{\boldsymbol{\Theta}^*}| \lesssim \tau_{\boldsymbol{\alpha},k} H_p \max\left\{kp\sqrt{\frac{L}{n}}, (kp)^{3/2}\frac{\sqrt{|\mathcal{I}|}}{n}\right\}.$$

### 3.5   Inference for Fixed $k$ and $p$ Under the MoM Framework

We now focus our attention back to the classical setting where the numbers of clusters and features remain fixed. To show $\widehat{\boldsymbol{\Theta}}_n^{(\text{MoM})}$ is consistent for $\boldsymbol{\Theta}^*$, we need to impose conditions such that the RHS of the bounds presented in Corollary 3.5 decrease to 0 as $n \to \infty$. We state the required conditions as follows.

**A7.** *The number of partitions $L = o(n)$, and $L \to \infty$ as $n \to \infty$.*

These conditions are natural: as $n$ grows, so too must $L$ in order to maintain a proportion of outlier-free partitions. On the other hand, $L$ must grow slowly relative to $n$ to ensure each partition can be assigned sufficient numbers of datapoints. We note that A7 implies $|\mathcal{O}| = o(n)$, an intuitive and standard condition [30, 46, 41] as outliers should be few by definition.

In the following corollary, we focus on the squared Euclidean distance for center-based clustering under the MoM framework. We show that the (global) estimates obtained from MoM $k$-means, MoM power $k$-means etc. are consistent. We stress that the obtained convergence rate, as a function of $n$, $L$ and $|\mathcal{I}|$, does *not* depend on the choice of $\Psi_{\boldsymbol{\alpha}}(\cdot)$, as long as it satisfies A2, i.e. is Lipschitz. In particular, replacing $\Psi_{\boldsymbol{\alpha}}(\cdot)$ with $\min_{1 \leq j \leq k} x_j$, $M_s(\boldsymbol{x})$ and $(\sum_{j=1}^{k} x_j^{-1})^{-1}$, the rate in Corollary 3.6 applies to robustified MoM versions of $k$-means, power $k$-means and $k$-harmonic means alike.

**Corollary 3.6.** *Suppose $\phi(\boldsymbol{u}) = \|\boldsymbol{u}\|_2^2$ and $\Psi_{\boldsymbol{\alpha}}(\cdot)$ satisfy A2. Then under A2—3 and A5—7,*

$$|Pf_{\widehat{\boldsymbol{\Theta}}_n^{(MoM)}} - Pf_{\boldsymbol{\Theta}^*}| = O_P\left(\max\left\{L^{1/2}n^{-1/2}, n^{-1}\sqrt{|\mathcal{I}|}\right\}\right)$$

*and $Pf_{\widehat{\boldsymbol{\Theta}}_n^{(MoM)}} \xrightarrow{P} Pf_{\boldsymbol{\Theta}^*}$. Moreover, whenever A4 additionally holds, we have $\widehat{\boldsymbol{\Theta}}_n^{(MoM)} \xrightarrow{P} \boldsymbol{\Theta}^*$.*

**Remark**   These results imply that any MoM center-based algorithm under our paradigm admits a convergence rate of $O\left(\max\left\{L^{1/2}n^{-1/2}, n^{-1}\sqrt{|\mathcal{I}|}\right\}\right)$, when equipped with squared Euclidean distance. Note that as $L \geq 1$, $\max\left\{L^{1/2}n^{-1/2}, n^{-1}\sqrt{|\mathcal{I}|}\right\} = \Omega\left(n^{-1/2}\right)$. Thus, the convergence

rates for MoM variants in our framework are generally slower than their ERM counterparts, for which the rate is $O(n^{-1/2})$. This is unsurprising as MoM operates on outlier-contaminated data; there is "no free lunch" in trading off robustness for rate of convergence. However, if the number of partitions $L$ grows slowly relative to $n$ (say, $L = O(\log n)$ so that $|\mathcal{O}| = O(\log n)$), then the convergence rates for MoM estimation become comparable to the ERM counterparts at $\widetilde{O}(n^{-1/2})$.

## 4 Empirical Studies and Performance

To validate and assess our proposed framework, we now turn to an empirical comparison of the proposed and peer clustering methods. We evaluate clustering quality under the Adjusted Rand Index (ARI) [23], with values ranging between 0 and 1 and 1 denoting a perfect match with the ground truth. Though it is not feasible to exhaustively survey center-based clustering methods, we consider a broad range of competitors, comparing to $k$-means [35], Partition Around Medoids (PAM) [25], $k$-medians [24], Robust kmeans++ (RKMpp) [18], Robust Continuous Clustering (RCC) [45], Bregman Trimmed $k$-means (BTKM) [21], MoM $k$-means (MOMKM) [26] and a novel MoM variant of power $k$-means (MOMPKM) implied under the proposed framework.

We consider two thorough simulated experiments below, with additional simulations and large-scale real data results in the Supplement. While the extended comparisons are omitted for space considerations, they convey the same trends as the studies below. In all settings, we generate data in $p = 5$ dimensions, varying the number of clusters and the outlier percentages. True centers are spaced uniformly on a grid with $\theta_k = \frac{k-1}{10}$ and observations are drawn from Gaussians around their ground truth centers with variance 0.1. Because we generate Gaussian data, we focus here on the Euclidean case, and do not consider other Bregman divergences in the present empirical study. In all experiments, we take the number of partitions $L$ to be roughly double the number of outliers, and set the hyperparameters $\eta$ and $\alpha$ to be 1.02 and 1 respectively by default. Results are averaged over 20 random restarts, and all competitors are initialized and tuned according to the standard implementations described in their original papers.

**Experiment 1: increasing the number of clusters**    The first experiment assesses performance as the number of true clusters grows, while keeping the proportion of outliers fixed at 25%. Datasets are generated with the number of clusters $k$ varying between 3 and 100. For each setting, we create balanced clusters drawing 30 points from each true center. The 25% outliers are generated from a uniform distribution with support on the range of the inlying observations. We repeat this data generation process 30 times under each parameter setting.

The average ARI values at convergence, plotted against the number of clusters along with error bars ($\pm$ standard deviations), are shown in the left panel of Figure 1. We see that the robustified version of power $k$-means implied by our framework (labeled MOMPKM) achieves the best performance here. This may be unsurprising as the recent power $k$-means method was shown to significantly reduce sensitivity to local minima (which tend to increase with $k$), while MoM further protects the algorithm from outlier influence.

**Experiment 2: increasing outlier percentage**    Following the same data generation process in Experiment 1, we now fix $k = 20$ while varying the outlier percentage from 0% to 50%. For each parameter setting, we again replicate the experiment 30 times. Average ARI values comparing the inlying observations to their ground truths are plotted in the right panel of Figure 1. Not only does this study convey similar trends as Experiment 1, but we see that competing methods continue to deteriorate with increasing outliers as one might expect, while MOMPKM remarkably remains relatively stable.

We see that the ERM-based methods such as BTKM, MOMKM, and PAM struggle when there is large number of clusters. Similarly, methods such as $k$-medians and RKMpp often stop short at poor local optima, which is quickly exacerbated by outliers despite initialization via clever seeding techniques. These phenomena are consistent with what has been reported in the literature for their non-robust counterparts [56, 17]. Overall, the empirical study suggests that a robust version of power $k$-means clustering under the MoM framework shows promise to handle several data challenges at once, in line with our theoretical analysis.

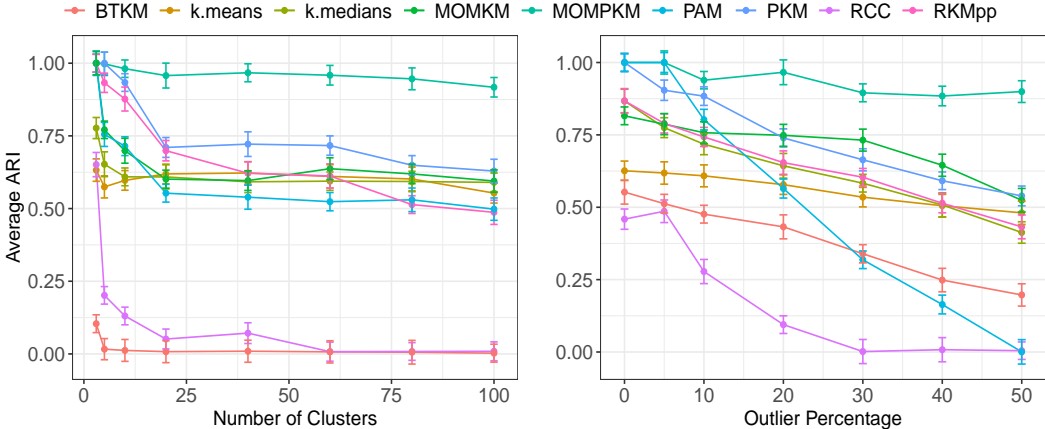

Figure 1: Average ARI values, along with error bars ($\pm$sd), comparing the peer algorithms in Experiments 1 (left) and 2 (right). MOMPKM remains relatively stable even when $k$ or outlier percentages increase, maintaining the best performance among peer methods.

# 5 Discussion

In this paper, we proposed a paradigm for center-based clustering that unifies a suite of center-based clustering algorithms. Under this view, developed a simple yet efficient framework for robust versions of such algorithms by appealing to the Median of Means (MoM) philosophy. Using gradient-based methods, the MoM objectives can be solved with the same per-iteration complexity of Lloyd's $k$-means algorithm, largely retaining its simplicity. Importantly, we derive a thorough analysis of the statistical properties and convergence rates by establishing uniform concentration bounds under i.i.d. sampling of the data. These novel theoretical contributions demonstrate how arguments utilizing Rademacher complexities and Dudley's chaining arguments can be leveraged in the robust clustering context. As a result, we are able to obtain error rates that do not require asymptotic assumptions, nor restrictions on the relation between $n$ and $p$. These findings recover asymptotic results such as strong consistency and $\sqrt{n}$-consistency under classical assumptions.

As shown in the paper, the robustness of MoM estimators comes at the cost of slower convergence rates compared to their ERM counterparts. We emphasize that there is no "median-of-means magic", and that the efficacy of MoM depends on the interplay between the partitions and the outliers. If the number of partitions circumvents the impact of the outliers, the performance of MoM clustering estimates under our framework scales with the block size $b$ as $1/\sqrt{b} = \sqrt{L/n}$. Since $L$ can be chosen to be approximately $2|\mathcal{O}|$, the obtained error rate is roughly $O(\sqrt{|\mathcal{O}|/n})$. If $|\mathcal{O}|$ scales proportionally with $n$, however, the error bound of $O(\sqrt{|\mathcal{O}|/n})$ becomes meaningless. For our consistency results to hold, it is crucial that $|\mathcal{O}| = o(n)$, which in turn allows us to choose $L$ that satisfies condition A7. If $|\mathcal{O}| = O(n^\beta)$, for some $0 < \beta < 1$, the error rate is $O(n^{(\beta-1)/2})$.

This suggests possible future research directions in improving these ERM rates via finding so called "fast rates" under additional assumptions [11, 55]. Moreover, it will be fruitful to extend the results for noise distributions that satisfy only moment conditions. Recent works in convex clustering [47, 16] have considered sub-Gaussian models to obtain error rates. The work by [10] and recent work by [26] obtain similar error rates under finite second-moment conditions using assumptions that the cluster centroids $\widehat{\Theta}_n$ are bounded, and it may be possible to extend our approach using local Rademacher complexities [7] to relax the bounded support assumption. One can also seek lower bounds on the approximation error or can explore high-dimensional robust center-based clustering under the proposed paradigm. Finally, we have not explored the implementation and empirical performance of Bregman versions of our MoM estimator, for instance with application to data arising from mixtures of exponential families other than the Gaussian case. These interesting directions remain open for future work.

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

## Funding Transparency Statement

The authors did not receive any third-party funding or third-party support during the last 36 months prior to this submission toward pursuing this work. The authors have no financial relationships with entities that could potentially be perceived to influence the submitted work.

