# Supplement to "Uniform Concentration Bounds toward a Unified Framework for Robust Clustering"

**Debolina Paul**[*]
Department of Statistics
Stanford University
deblinap@stanford.edu

**Saptarshi Chakraborty**[*]
Department of Statistics
UC Berkeley
saptarshic@berkeley.edu

**Swagatam Das**
Electronics and Communications Sciences Unit
Indian Statistical Institute
swagatam.das@isical.ac.in

**Jason Xu**[‡]
Department of Statistical Science
Duke University
jason.q.xu@duke.edu

## A  Proofs of Lemmas

For the theoretical exposition, we first establish the following Lemmas. Lemma A.1 proves that the derivative of the function $\phi$ is bounded in the $\ell_2$-norm when the domain is restricted to the support of $P$.

**Lemma A.1.** *Under A3, $\|\nabla\phi(\boldsymbol{x})\|_2 \le H_p M \sqrt{p}$, for all $\boldsymbol{x} \in [-M, M]^p$.*

*Proof.* From A3, we observe that

$$\|\nabla\phi(\boldsymbol{x}) - \nabla\phi(\boldsymbol{0})\|_2 \le H_p\|\boldsymbol{x}\|_2$$
$$\implies \|\nabla\phi(\boldsymbol{x})\|_2 \le H_p\|\boldsymbol{x}\|_2 \le H_p M \sqrt{p}.$$

$\square$

Lemma A.2 essentially proves that the function $\phi$ is Lipschitz with Lipschitz constant $H_p M \sqrt{p}$ on $[-M, M]^p$.

**Lemma A.2.** *Under A3, for all $\boldsymbol{x}, \boldsymbol{y} \in [-M, M]^p$, $\phi(\cdot)$ is $2H_p M \sqrt{p}$-Lipschitz, i.e.*

$$|\phi(\boldsymbol{x}) - \phi(\boldsymbol{y})| \le H_p M \sqrt{p}\|\boldsymbol{x} - \boldsymbol{y}\|_2.$$

*Proof.* From the mean value theorem,

$$\phi(\boldsymbol{x}) - \phi(\boldsymbol{y}) = \langle \nabla\phi(\boldsymbol{\xi}), \boldsymbol{x} - \boldsymbol{y}\rangle,$$

for some $\boldsymbol{\xi}$ in the convex combinations of $\boldsymbol{x}$ and $\boldsymbol{y}$. Clearly, $\boldsymbol{\xi} \in [-M, M]^p$, due to the convexity of $[-M, M]^p$. Now by the Cauchy-Schwartz inequality and Lemma A.1,

$$|\phi(\boldsymbol{x}) - \phi(\boldsymbol{y})| \le \|\nabla\phi(\boldsymbol{\xi})\|_2\|\boldsymbol{x} - \boldsymbol{y}\|_2 \le H_p M \sqrt{p}\|\boldsymbol{x} - \boldsymbol{y}\|_2.$$

$\square$

Lemma A.3 proves that the function $f_{\boldsymbol{\Theta}}$, as a function of $\boldsymbol{\Theta}$, is Lipschitz with respect to the $\|\cdot\|_\infty$ norm.

---

[*]Joint first authors contributed equally   [‡] Corresponding author

35th Conference on Neural Information Processing Systems (NeurIPS 2021).

**Lemma A.3.** *For any* $\boldsymbol{\Theta}, \boldsymbol{\Theta}' \in [-M, M]^p,$

$$\|f_{\boldsymbol{\Theta}} - f_{\boldsymbol{\Theta}'}\|_\infty \le 4\tau_{\boldsymbol{\alpha}, k} H_p M \sqrt{p} \sum_{j=1}^k \|\boldsymbol{\theta}'_j - \boldsymbol{\theta}_j\|_2.$$

*Here,* $\boldsymbol{\Theta} = [\boldsymbol{\theta}_1^\top, \ldots, \boldsymbol{\theta}_k^\top]^\top$ *and* $\boldsymbol{\Theta} = [\boldsymbol{\theta}_1'^\top, \ldots, \boldsymbol{\theta}_k'^\top]^\top.$

*Proof.*

$\|f_{\boldsymbol{\Theta}} - f_{\boldsymbol{\Theta}'}\|_\infty$

$= \displaystyle\sup_{\boldsymbol{x} \in [-M,M]^p} \left| \Psi_{\boldsymbol{\alpha}}(d_\phi(\boldsymbol{x}, \boldsymbol{\theta}_1), \ldots, d_\phi(\boldsymbol{x}, \boldsymbol{\theta}_k)) - \Psi_{\boldsymbol{\alpha}}(d_\phi(\boldsymbol{x}, \boldsymbol{\theta}_1'), \ldots, d_\phi(\boldsymbol{x}, \boldsymbol{\theta}_k')) \right|$

$\le \tau_{\boldsymbol{\alpha}, k} \displaystyle\sum_{j=1}^k |d_\phi(\boldsymbol{x}, \boldsymbol{\theta}_j) - d_\phi(\boldsymbol{x}, \boldsymbol{\theta}_j')|$

$= \tau_{\boldsymbol{\alpha}, k} \displaystyle\sum_{j=1}^k |\phi(\boldsymbol{\theta}_j') - \phi(\boldsymbol{\theta}_j) + \langle \nabla\phi(\boldsymbol{\theta}_j'), \boldsymbol{x} - \boldsymbol{\theta}_j' \rangle - \langle \nabla\phi(\boldsymbol{\theta}_j), \boldsymbol{x} - \boldsymbol{\theta}_j \rangle |$

$= \tau_{\boldsymbol{\alpha}, k} \displaystyle\sum_{j=1}^k |\phi(\boldsymbol{\theta}_j') - \phi(\boldsymbol{\theta}_j) + \langle \nabla\phi(\boldsymbol{\theta}_j') - \nabla\phi(\boldsymbol{\theta}_j), \boldsymbol{x} - \boldsymbol{\theta}_j' \rangle + \langle \nabla\phi(\boldsymbol{\theta}_j), \boldsymbol{\theta}_j - \boldsymbol{\theta}_j' \rangle |$

$\le \tau_{\boldsymbol{\alpha}, k} \displaystyle\sum_{j=1}^k \left( |\phi(\boldsymbol{\theta}_j') - \phi(\boldsymbol{\theta}_j)| + |\langle \nabla\phi(\boldsymbol{\theta}_j') - \nabla\phi(\boldsymbol{\theta}_j), \boldsymbol{x} - \boldsymbol{\theta}_j' \rangle| + |\langle \nabla\phi(\boldsymbol{\theta}_j), \boldsymbol{\theta}_j - \boldsymbol{\theta}_j' \rangle| \right)$

$\le \tau_{\boldsymbol{\alpha}, k} \displaystyle\sum_{j=1}^k \left( H_p M \sqrt{p} \|\boldsymbol{\theta}_j' - \boldsymbol{\theta}_j\|_2 + \|\nabla\phi(\boldsymbol{\theta}_j') - \nabla\phi(\boldsymbol{\theta}_j)\|_2 \|\boldsymbol{x} - \boldsymbol{\theta}_j'\|_2 + \|\nabla\phi(\boldsymbol{\theta}_j)\|_2 \|\boldsymbol{\theta}_j - \boldsymbol{\theta}_j'\|_2 \right)$

$\le \tau_{\boldsymbol{\alpha}, k} \displaystyle\sum_{j=1}^k \left( H_p M \sqrt{p} \|\boldsymbol{\theta}_j' - \boldsymbol{\theta}_j\|_2 + H_p \|\boldsymbol{\theta}_j' - \boldsymbol{\theta}_j\|_2 \times 2\sqrt{p} M + H_p M \sqrt{p} \|\boldsymbol{\theta}_j - \boldsymbol{\theta}_j'\|_2 \right)$

$\le 4\tau_{\boldsymbol{\alpha}, k} H_p M \sqrt{p} \displaystyle\sum_{j=1}^k \|\boldsymbol{\theta}_j' - \boldsymbol{\theta}_j\|_2$

$\square$

# B  Proofs from Section 3

## B.1  Proof of Lemma 3.1

*Proof.* Let $J(\boldsymbol{x}) = d_\phi(\boldsymbol{x}, \boldsymbol{\theta})$. Since $P_{\mathcal{C}}(\boldsymbol{\theta})$ minimizes $J(\cdot)$ over $\mathcal{C}$, there exists a subgradient $\boldsymbol{d} \in \partial J(P_{\mathcal{C}}(\boldsymbol{\theta}))$ such that

$$\langle \boldsymbol{d}, \boldsymbol{x} - P_{\mathcal{C}}(\boldsymbol{\theta}) \rangle \ge 0. \tag{1}$$

We note that $J(P_{\mathcal{C}}(\boldsymbol{\theta})) = \{\nabla\phi(P_{\mathcal{C}}(\boldsymbol{\theta})) - \nabla\phi(\boldsymbol{\theta})\}$. Thus, from equation (1),

$$\langle \nabla\phi(P_{\mathcal{C}}(\boldsymbol{\theta})) - \nabla\phi(\boldsymbol{\theta}), \boldsymbol{x} - P_{\mathcal{C}}(\boldsymbol{\theta}) \rangle \ge 0. \tag{2}$$

We now observe that,

$$d_\phi(\boldsymbol{x}, \boldsymbol{\theta}) - d_\phi(\boldsymbol{x}, P_{\mathcal{C}}(\boldsymbol{\theta})) - d_\phi(P_{\mathcal{C}}(\boldsymbol{\theta}), \boldsymbol{\theta}) = \langle \nabla\phi(P_{\mathcal{C}}(\boldsymbol{\theta})) - \nabla\phi(\boldsymbol{\theta}), \boldsymbol{x} - P_{\mathcal{C}}(\boldsymbol{\theta}) \rangle \ge 0.$$

Hence the result. $\square$

## B.2  Proof of Lemma 3.2

*Proof.* Suppose $\boldsymbol{\Theta} = \{\boldsymbol{\theta}_1, \ldots, \boldsymbol{\theta}_k\}$. We take $\mathcal{C} = [-M, M]^{k \times p}$ and $\boldsymbol{\Theta}' = \{P_{\mathcal{C}}(\boldsymbol{\theta}_1), \ldots, P_{\mathcal{C}}(\boldsymbol{\theta}_k)\}$. Clearly $\mathcal{C}$ is a convex set. Thus, from Lemma 3.1, we observe that

$$d_\phi(\boldsymbol{x}, \boldsymbol{\theta}_j) \ge d_\phi(\boldsymbol{x}, P_{\mathcal{C}}(\boldsymbol{\theta}_j)) + d_\phi(P_{\mathcal{C}}(\boldsymbol{\theta}_j), \boldsymbol{\theta}_j) \ge d_\phi(\boldsymbol{x}, P_{\mathcal{C}}(\boldsymbol{\theta}_j)) \quad \forall j = 1, \ldots, k.$$

$$\implies \Psi_{\boldsymbol{\alpha}}\left(d_\phi(\boldsymbol{x}, P_{\mathcal{C}}(\boldsymbol{\theta}_1)), \ldots, d_\phi(\boldsymbol{x}, P_{\mathcal{C}}(\boldsymbol{\theta}_k))\right) \le \Psi_{\boldsymbol{\alpha}}\left(d_\phi(\boldsymbol{x}, \boldsymbol{\theta}_1), \ldots, d_\phi(\boldsymbol{x}, \boldsymbol{\theta}_k)\right)$$

$$\implies \int \Psi_{\boldsymbol{\alpha}}\left(d_\phi(\boldsymbol{x}, P_{\mathcal{C}}(\boldsymbol{\theta}_1)), \ldots, d_\phi(\boldsymbol{x}, P_{\mathcal{C}}(\boldsymbol{\theta}_k))\right) dQ \le \int \Psi_{\boldsymbol{\alpha}}\left(d_\phi(\boldsymbol{x}, \boldsymbol{\theta}_1), \ldots, d_\phi(\boldsymbol{x}, \boldsymbol{\theta}_k)\right) dQ$$

$$\implies Qf_{\boldsymbol{\Theta}'} \le Qf_{\boldsymbol{\Theta}}$$

$\square$

## B.3 Proof of Lemma 3.3

*Proof.* We first divide the set $[-M, M]$ into a small bins, each with size $\epsilon$. Denote $\gamma_i = -M + i\epsilon$, for $i = 1, \ldots, \lfloor \frac{2M}{\epsilon} \rfloor$, and let $\Gamma_\epsilon = \left\{\gamma_i \,\|\, i \in \{1, \ldots, \lfloor \frac{2M}{\epsilon} \rfloor\}\right\}$. If $\epsilon > 2M$, we take $\Gamma_\epsilon = \{0\}$. Clearly, $|\Gamma_\epsilon| = \max\{\lfloor \frac{2M}{\epsilon} \rfloor, 1\}$. From the construction of $\Gamma_\epsilon$, for all $x \in [-M, M]$, there exists $i \in [|\Gamma_\epsilon|]$, such that, $|x - \gamma_i| \le \epsilon$. We take $\epsilon = (4\tau_{\boldsymbol{\alpha},k} H_p Mkp)^{-1} \delta$. We define

$$\boldsymbol{\Theta}_\delta = \{\boldsymbol{\Theta} = ((\theta_{i\ell})) : \theta_{i\ell} \in \Gamma_\epsilon\}.$$

Then immediately we see

$$|\boldsymbol{\Theta}_\delta| = \left(\max\left\{\left\lfloor \frac{2M}{\epsilon} \right\rfloor, 1\right\}\right)^{kp}.$$

For any $\boldsymbol{\Theta} \in [-M, M]^p$, we can construct $\boldsymbol{\Theta}' \in \boldsymbol{\Theta}_\delta$, such that, $|\theta_{i\ell} - \theta'_{i\ell}| \le \epsilon$. From Lemma A.3, we observe that,

$$\|f_{\boldsymbol{\Theta}} - f_{\boldsymbol{\Theta}'}\|_\infty \le 4\tau_{\boldsymbol{\alpha},k} H_p M \sqrt{p} \sum_{j=1}^k \|\boldsymbol{\theta}'_j - \boldsymbol{\theta}_j\|_2.$$

$$\le 4\tau_{\boldsymbol{\alpha},k} H_p M \sqrt{p} k \sqrt{p} \epsilon$$

$$= 4\tau_{\boldsymbol{\alpha},k} H_p Mkp\epsilon$$

$$= \delta.$$

Thus, $\mathcal{F}_\delta = \{f_{\boldsymbol{\Theta}} : \boldsymbol{\Theta} \in \boldsymbol{\Theta}_\delta\}$ constitutes a $\delta$-cover of $\mathcal{F}$ under the $\|\cdot\|_\infty$ norm. Hence,

$$N(\delta; \mathcal{F}, \|\cdot\|_\infty) \le |\mathcal{F}_\delta| \le |\boldsymbol{\Theta}_\delta| = \left(\max\left\{\left\lfloor \frac{2M}{\epsilon} \right\rfloor, 1\right\}\right)^{kp}$$

$$= \left(\max\left\{\left\lfloor \frac{8M^2 \tau_{\boldsymbol{\alpha},k} H_p kp}{\delta} \right\rfloor, 1\right\}\right)^{kp}.$$

$\square$

## B.4 Proof of Lemma 3.4

*Proof.* From Lemma A.3, we observe that,

$$\mathrm{diam}(\mathcal{F}) = \sup_{\boldsymbol{\Theta}, \boldsymbol{\Theta}' \in [-M, M]^{k \times p}} \|f_{\boldsymbol{\Theta}} - f_{\boldsymbol{\Theta}'}\|_\infty$$

$$\le 4H_p M \sqrt{p} \tau_{\boldsymbol{\alpha},k} \sup_{\boldsymbol{\Theta}, \boldsymbol{\Theta}' \in [-M, M]^{k \times p}} \sum_{j=1}^k \|\boldsymbol{\theta}'_j - \boldsymbol{\theta}_j\|_2$$

$$\le 4H_p M \sqrt{p} \tau_{\boldsymbol{\alpha},k} \times 2kM\sqrt{p}$$

$$= 8\tau_{\boldsymbol{\alpha},k} H_p M^2 kp.$$

$\square$

## B.5 Proof of Lemma 3.5

*Proof.* From the non-negativity of $\Psi_{\boldsymbol{\alpha}}(\cdot)$, we get, $\Psi_{\boldsymbol{\alpha}}(d_\phi(\boldsymbol{x}, \boldsymbol{\theta}_1), \ldots, d_\phi(\boldsymbol{x}, \boldsymbol{\theta}_k)) \ge 0$, for any $\boldsymbol{x} \in [-M, M]^p$ and $\boldsymbol{\Theta} \in [-M, M]^{k \times p}$. For any $\boldsymbol{\beta} \in \mathbb{R}_{\ge 0}^k$, from A3, we get,

$$\Psi_{\boldsymbol{\alpha}}(\boldsymbol{\beta}) = |\Psi_{\boldsymbol{\alpha}}(\boldsymbol{\beta}) - \Psi_{\boldsymbol{\alpha}}(\boldsymbol{0})| \le \tau_{\boldsymbol{\alpha},k} \|\boldsymbol{\beta} - \boldsymbol{0}\|_1 = \|\boldsymbol{\beta}\|_1.$$

Taking $\boldsymbol{\beta} = (d_\phi(\boldsymbol{x}, \boldsymbol{\theta}_1), \ldots, d_\phi(\boldsymbol{x}, \boldsymbol{\theta}_k))^\top$, we get,

$$\Psi_{\boldsymbol{\alpha}}(d_\phi(\boldsymbol{x}, \boldsymbol{\theta}_1), \ldots, d_\phi(\boldsymbol{x}, \boldsymbol{\theta}_k))$$

$$\leq \tau_{\boldsymbol{\alpha},k} \sum_{j=1}^{k} d_\phi(\boldsymbol{x}, \boldsymbol{\theta}_j)$$

$$= \tau_{\boldsymbol{\alpha},k} \sum_{j=1}^{k} \left(\phi(\boldsymbol{x}) - \phi(\boldsymbol{\theta}_j) - \langle \nabla\phi(\boldsymbol{\theta}_j), \boldsymbol{x} - \boldsymbol{\theta}_j \rangle\right)$$

$$\leq \tau_{\boldsymbol{\alpha},k} \sum_{j=1}^{k} \left(|\phi(\boldsymbol{x}) - \phi(\boldsymbol{\theta}_j)| + |\langle \nabla\phi(\boldsymbol{\theta}_j), \boldsymbol{x} - \boldsymbol{\theta}_j \rangle|\right)$$

$$\leq \tau_{\boldsymbol{\alpha},k} \sum_{j=1}^{k} \left(H_p M \sqrt{p} \|\boldsymbol{x} - \boldsymbol{\theta}_j\|_2 + \|\nabla\phi(\boldsymbol{\theta}_j)\|_2 \|\boldsymbol{x} - \boldsymbol{\theta}_j\|_2\right) \tag{3}$$

$$\leq 2\tau_{\boldsymbol{\alpha},k} H_p M \sqrt{p} \sum_{j=1}^{k} \|\boldsymbol{x} - \boldsymbol{\theta}_j\|_2 \tag{4}$$

$$\leq 4\tau_{\boldsymbol{\alpha},k} H_p M^2 p k.$$

Here inequality (3) follows from Cauchy-Schwartz inequality and Lemma A.2. Inequality (4) follows from Lemma A.1. $\qquad\square$

## B.6   Proof of Theorem 3.1

*Proof.* Let $\Delta = 8 H_p M^2 k^{1-1/s} p$. We construct a decreasing sequence $\{\delta_i\}_{i \in \mathbb{N}}$ as follows. Take $\delta_1 := \text{diam}(\mathcal{F}) = \Delta$ (the last equality follows from Lemma 3.4) and $\delta_{i+1} = \frac{1}{2}\delta_i$. Let $\mathcal{F}_i$ be a minimal $\delta_i$ cover of $\mathcal{F}$, i.e. $|\mathcal{F}_i| = N(\delta_i; \mathcal{F}, \|\cdot\|_\infty)$. Now denote $f_i$ to be the closest element of $f$ in $\mathcal{F}_i$ (with ties broken arbitrarily). We can thus write,

$$\mathbb{E} \sup_{f \in \mathcal{F}} \frac{1}{n} \sum_{i=1}^{n} \epsilon_i f(\boldsymbol{X}_i) \leq \xi_1 + \xi_2 + \xi_3,$$

where

$$\xi_1 = \mathbb{E} \sup_{f \in \mathcal{F}} \frac{1}{n} \sum_{i=1}^{n} \epsilon_i (f(\boldsymbol{X}_i) - f_m(\boldsymbol{X}_i)), \tag{5}$$

$$\xi_2 = \sum_{j=1}^{m-1} \mathbb{E} \sup_{f \in \mathcal{F}} \frac{1}{n} \sum_{i=1}^{n} \epsilon_i (f_{j+1}(\boldsymbol{X}_i) - f_j(\boldsymbol{X}_i)), \tag{6}$$

$$\xi_3 = \mathbb{E} \sup_{f \in \mathcal{F}} \frac{1}{n} \sum_{i=1}^{n} \epsilon_i f_1(\boldsymbol{X}_i). \tag{7}$$

Since we can pick $f_1$ arbitrarily from $\mathcal{F}$ (as $\delta_1 = \text{diam}(\mathcal{F})$), $\xi_3 = 0$. To bound $\xi_1$, we observe that,

$$\xi_1 = \mathbb{E} \sup_{f \in \mathcal{F}} \frac{1}{n} \sum_{i=1}^{n} \epsilon_i (f(\boldsymbol{X}_i) - f_m(\boldsymbol{X}_i)) \leq \mathbb{E} \sup_{f \in \mathcal{F}} \frac{1}{n} \sqrt{\left(\sum_{i=1}^{n} \epsilon_i^2\right)\left(\sum_{i=1}^{n} (f(\boldsymbol{X}_i) - f_m(\boldsymbol{X}_i))^2\right)} \leq \delta_m$$

To bound $\xi_2$, we observe that,

$$\|f_{j+1} - f_j\|_\infty \leq \|f_{j+1} - f\|_\infty + \|f - f_j\|_\infty \leq \delta_{j+1} + \delta_j.$$

Now appealing to Massart's lemma [4], we get,

$$\mathbb{E} \sup_{f \in \mathcal{F}} \frac{1}{n} \sum_{i=1}^{n} \epsilon_i (f_{j+1}(\boldsymbol{X}_i) - f_j(\boldsymbol{X}_i)) \leq \frac{(\delta_{j+1} + \delta_j)\sqrt{2 \log\left(N(\delta_j; \mathcal{F}, \|\cdot\|_\infty) N(\delta_{j+1}; \mathcal{F}, \|\cdot\|_\infty)\right)}}{\sqrt{n}}$$

$$\leq \frac{2(\delta_{j+1} + \delta_j)\sqrt{\log N(\delta_{j+1}; \mathcal{F}, \|\cdot\|_\infty)}}{\sqrt{n}}$$

Thus,

$$\xi_2 = \sum_{j=1}^{m-1} \mathbb{E} \sup_{f \in \mathcal{F}} \frac{1}{n} \sum_{i=1}^{n} \epsilon_i(f_{j+1}(\boldsymbol{X}_i) - f_j(\boldsymbol{X}_i)) \leq \sum_{j=1}^{m-1} \frac{2(\delta_{j+1} + \delta_j)\sqrt{\log N(\delta_{j+1}; \mathcal{F}, \|\cdot\|_\infty)}}{\sqrt{n}}$$

Combining the bounds on $\xi_1$, $\xi_2$ and $\xi_3$, we get,

$$\mathbb{E} \sup_{f \in \mathcal{F}} \frac{1}{n} \sum_{i=1}^{n} \epsilon_i f(\boldsymbol{X}_i) \leq \delta_m + \frac{2}{\sqrt{n}} \sum_{j=1}^{m-1} (\delta_{j+1} + \delta_j)\sqrt{\log N(\delta_{j+1}; \mathcal{F}, \|\cdot\|_\infty)}. \tag{8}$$

From the construction of $\{\delta_i\}_{i \geq 1}$, we know, $\delta_{j+1} + \delta_j = 6(\delta_{j+1} - \delta_{j+2})$. Hence from equation 8, we get,

$$\mathbb{E} \sup_{f \in \mathcal{F}} \frac{1}{n} \sum_{i=1}^{n} \epsilon_i f(\boldsymbol{X}_i) \leq \delta_m + \frac{2}{\sqrt{n}} \sum_{j=1}^{m-1} (\delta_{j+1} + \delta_j)\sqrt{\log N(\delta_{j+1}; \mathcal{F}, \|\cdot\|_\infty)}$$

$$= \delta_m + \frac{12}{\sqrt{n}} \sum_{j=1}^{m-1} (\delta_{j+1} - \delta_{j+2})\sqrt{\log N(\delta_{j+1}; \mathcal{F}, \|\cdot\|_\infty)}$$

$$\leq \delta_m + \frac{12}{\sqrt{n}} \int_{\delta_{m+1}}^{\delta_2} \sqrt{\log N(\epsilon; \mathcal{F}, \|\cdot\|_\infty)} d\epsilon$$

Taking limits as $m \to \infty$ in the above equation, we get,

$$\mathbb{E} \sup_{f \in \mathcal{F}} \frac{1}{n} \sum_{i=1}^{n} \epsilon_i f(\boldsymbol{X}_i) \leq \frac{12}{\sqrt{n}} \int_{0}^{\Delta} \sqrt{\log N(\epsilon; \mathcal{F}, \|\cdot\|_\infty)} d\epsilon.$$

From Lemma 3.3, plugging in the value of $N(\epsilon; \mathcal{F}, \|\cdot\|_\infty)$, we get,

$$\mathcal{R}_n(\mathcal{F}) \leq \frac{12}{\sqrt{n}} \int_{0}^{\Delta} \sqrt{\log N(\epsilon; \mathcal{F}, \|\cdot\|_\infty)} d\epsilon$$

$$\leq \frac{12}{\sqrt{n}} \int_{0}^{\Delta} \sqrt{kp \log\left(\max\left\{\frac{\Delta}{\epsilon}, 1\right\}\right)} d\epsilon$$

$$= \frac{12}{\sqrt{n}} \int_{0}^{\Delta} \sqrt{kp \log\left(\frac{\Delta}{\epsilon}\right)} d\epsilon$$

$$= 12\sqrt{\frac{kp}{n}} \Delta \int_{0}^{\infty} 2t^2 e^{-t^2} dt$$

$$= 12\sqrt{\frac{kp}{n}} \Delta \int_{0}^{\infty} u^{\frac{3}{2}-1} e^{-u} du$$

$$= 12\sqrt{\frac{kp}{n}} \Delta \Gamma(3/2)$$

$$= 6\sqrt{\frac{kp\pi}{n}} \times 8\tau_{\boldsymbol{\alpha},k} H_p M^2 kp$$

$$= 48\sqrt{\pi}\tau_{\boldsymbol{\alpha},k} H_p M^2 (kp)^{3/2} n^{-1/2}.$$

$\square$

## B.7 Proof of Theorem 3.2

*Proof.* From Lemma, 3.5, we observe that $\sup_{f \in \mathcal{F}} \|f\|_\infty \leq 4\tau_{\boldsymbol{\alpha},k} H_p M^2 pk$. Under assumption A1, we observe that, with probability at least $1 - \delta$,

$$\sup_{f \in \mathcal{F}} |P_n f - Pf| \leq 2\mathcal{R}_n(\mathcal{F}) + \sup_{f \in \mathcal{F}} \|f\|_\infty \sqrt{\frac{\log(2/\delta)}{2n}}$$

$$\leq 96\sqrt{\pi}\tau_{\boldsymbol{\alpha},k} H_p M^2 (kp)^{3/2} n^{-1/2} + 4\tau_{\boldsymbol{\alpha},k} H_p M^2 pk \sqrt{\frac{\log(2/\delta)}{2n}}. \tag{9}$$

Inequality (9) follows from appealing to Theorem 3.1 and observing that $\sup_{f\in\mathcal{F}}\|f\|_\infty \leq 4\tau_{\boldsymbol{\alpha},k}H_pM^2pk$. $\qquad\square$

## B.8 Proof of Theorem 3.3

*Proof.* (Proof of Strong consistency) We will first show $|Pf_{\widehat{\boldsymbol{\Theta}}_n} - Pf_{\boldsymbol{\Theta}^*}| \xrightarrow{a.s.} 0$. To show this let $C = \max\{192\sqrt{\pi}\tau_{\boldsymbol{\alpha},k}H_pM^2(kp)^{3/2}, 8\tau_{\boldsymbol{\alpha},k}H_pM^2pk\}$. Then from Theorem 3.2, we observe that with probability at least $1 - \delta$,

$$|Pf_{\widehat{\boldsymbol{\Theta}}_n} - Pf_{\boldsymbol{\Theta}^*}| \leq \frac{C}{\sqrt{n}} + C\sqrt{\frac{\log(2/\delta)}{2n}}. \tag{10}$$

Fix $\epsilon > 0$. If $n \geq 4C^2/\epsilon^2$ and $\delta = 2\exp\left(-\frac{n\epsilon^2}{2C^2}\right)$, the RHS of (10) becomes no bigger than $\epsilon$. Thus,

$$\mathbb{P}\left(|Pf_{\widehat{\boldsymbol{\Theta}}_n} - Pf_{\boldsymbol{\Theta}^*}| > \epsilon\right) \leq 2\exp\left(-\frac{n\epsilon^2}{2C^2}\right), \quad \forall n \geq 4C^2/\epsilon^2.$$

Since the series $\sum_{n=1}^{\infty}\exp\left(-\frac{n\epsilon^2}{2C^2}\right)$ is convergent from the above equation, so is $\mathbb{P}\left(|Pf_{\widehat{\boldsymbol{\Theta}}_n} - Pf_{\boldsymbol{\Theta}^*}| > \epsilon\right)$. Hence, $Pf_{\widehat{\boldsymbol{\Theta}}_n} \xrightarrow{a.s.} Pf_{\boldsymbol{\Theta}^*}$. Thus, for any $\epsilon > 0$, $Pf_{\widehat{\boldsymbol{\Theta}}_n} \leq pf_{\boldsymbol{\Theta}^*} + \epsilon$ almost surely w.r.t. $[P]$ for $n$ sufficiently large. From assumption A4, $\text{diss}(\widehat{\boldsymbol{\Theta}}_n, \boldsymbol{\Theta}^*) \leq \eta$, almost surely w.r.t. $[P]$, for any prefixed $\eta > 0$, and $n$ large. Thus, $\text{diss}(\widehat{\boldsymbol{\Theta}}_n, \boldsymbol{\Theta}^*) \xrightarrow{a.s.} 0$, which proves the result.

(Proof of $\sqrt{n}$-consistency) Fix $\delta \in (0, 1]$. From Theorem 3.2, with probability at least $1 - \delta$,

$$|Pf_{\widehat{\boldsymbol{\Theta}}_n} - Pf_{\boldsymbol{\Theta}^*}| \leq 192\sqrt{\pi}\tau_{\boldsymbol{\alpha},k}H_pM^2(kp)^{3/2}n^{-1/2} + 8\tau_{\boldsymbol{\alpha},k}H_pM^2pk\sqrt{\frac{\log(2/\delta)}{2n}} = O(n^{-1/2}).$$

Hence, $\sqrt{n}|Pf_{\widehat{\boldsymbol{\Theta}}_n} - Pf_{\boldsymbol{\Theta}^*}| = O(1)$ with probability at least $1 - \delta$. Thus, $\exists C_\delta$, such that $\mathbb{P}\left(\sqrt{n}|Pf_{\widehat{\boldsymbol{\Theta}}_n} - Pf_{\boldsymbol{\Theta}^*}| \leq C_\delta\right) \geq 1 - \delta$ for all $n$ large enough. Hence, $|Pf_{\widehat{\boldsymbol{\Theta}}_n} - Pf_{\boldsymbol{\Theta}^*}| = O_P(n^{-1/2})$. $\qquad\square$

# C   Proofs from Section 3.4

## C.1   Proof of Lemma 3.6

*Proof.* Suppose $\boldsymbol{\Theta} = \{\boldsymbol{\theta}_1, \ldots, \boldsymbol{\theta}_k\}$. We take $\mathcal{C} = [-M, M]^{k\times p}$ and $\boldsymbol{\Theta}' = \{P_{\mathcal{C}}(\boldsymbol{\theta}_1), \ldots, P_{\mathcal{C}}(\boldsymbol{\theta}_k)\}$. Clearly $\mathcal{C}$ is convex. Let $\mathcal{L} \subset \{1, \ldots, L\}$ be the set of all partitions which do not contain an outlier. Thus, from Lemma 3.1, we observe that

$$d_\phi(\boldsymbol{X}_i, \boldsymbol{\theta}_j) \geq d_\phi(\boldsymbol{X}_i, P_{\mathcal{C}}(\boldsymbol{\theta}_j)) + d_\phi(P_{\mathcal{C}}(\boldsymbol{\theta}_j), \boldsymbol{\theta}_j) \geq d_\phi(\boldsymbol{X}_i, P_{\mathcal{C}}(\boldsymbol{\theta}_j)) \, \forall j = 1, \ldots, k \text{ and } i \in \mathcal{I}$$
$$\implies \Psi_{\boldsymbol{\alpha}}\left(d_\phi(\boldsymbol{X}_i, P_{\mathcal{C}}(\boldsymbol{\theta}_1)), \ldots, d_\phi(\boldsymbol{X}, P_{\mathcal{C}}(\boldsymbol{\theta}_k))\right) \leq \Psi_{\boldsymbol{\alpha}}\left(d_\phi(\boldsymbol{X}_i, \boldsymbol{\theta}_1), \ldots, d_\phi(\boldsymbol{X}, \boldsymbol{\theta}_k)\right) \forall i \in \mathcal{I}$$
$$\implies \sum_{i\in B_\ell} \Psi_{\boldsymbol{\alpha}}\left(d_\phi(\boldsymbol{X}_i, P_{\mathcal{C}}(\boldsymbol{\theta}_1)), \ldots, d_\phi(\boldsymbol{X}, P_{\mathcal{C}}(\boldsymbol{\theta}_k))\right) \leq \sum_{i\in B_\ell} \Psi_{\boldsymbol{\alpha}}\left(d_\phi(\boldsymbol{X}_i, \boldsymbol{\theta}_1), \ldots, d_\phi(\boldsymbol{X}, \boldsymbol{\theta}_k)\right) \forall \ell \in \mathcal{L}$$
$$\implies \frac{1}{b}\sum_{i\in B_\ell} f_{\boldsymbol{\Theta}'}(\boldsymbol{X}_i) \leq \frac{1}{b}\sum_{i\in B_\ell} f_{\boldsymbol{\Theta}}(\boldsymbol{X}_i) \forall \ell \in \mathcal{L}$$

Now since $|\mathcal{L}| > |\mathcal{L}^C|$ (from assumption A6),

$$\text{Median}\left(\frac{1}{b}\sum_{i\in B_1} f_{\boldsymbol{\Theta}'}(\boldsymbol{X}_i), \ldots, \frac{1}{b}\sum_{i\in B_L} f_{\boldsymbol{\Theta}'}(\boldsymbol{X}_i)\right) \leq \text{Median}\left(\frac{1}{b}\sum_{i\in B_1} f_{\boldsymbol{\Theta}}(\boldsymbol{X}_i), \ldots, \frac{1}{b}\sum_{i\in B_L} f_{\boldsymbol{\Theta}}(\boldsymbol{X}_i)\right)$$
$$\implies \text{MoM}_L^n(\boldsymbol{\Theta}') \leq \text{MoM}_L^n(\boldsymbol{\Theta})$$

$\qquad\square$

## C.2  Proof of Theorem 3.4

*Proof.* For notational simplicity let $P_{B_\ell}$ denote the empirical distribution of $\{X_i\}_{i \in B_\ell}$. Suppose $\epsilon > 0$. We will first bound the probability of $\sup_{\Theta \in [-M,M]^{k \times p}} |\mathrm{MoM}_L^n(f_\Theta) - Pf_\Theta| > \epsilon$. To do so, we will individually bound the probabilities of the events

$$\sup_{\Theta \in [-M,M]^{k \times p}} (\mathrm{MoM}_L^n(f_\Theta) - Pf_\Theta) > \epsilon$$

and

$$\sup_{\Theta \in [-M,M]^{k \times p}} (Pf_\Theta - \mathrm{MoM}_L^n(f_\Theta)) > \epsilon.$$

We note that if

$$\sup_{\Theta \in [-M,M]^{k \times p}} \sum_{\ell=1}^{L} \mathbb{1}\left\{(P - P_{B_\ell})f_\Theta > \epsilon\right\} > \frac{L}{2},$$

then

$$\sup_{\Theta \in [-M,M]^{k \times p}} (Pf_\Theta - \mathrm{MoM}_L^n(f_\Theta)) > \epsilon.$$

Here again $\mathbb{1}\{\cdot\}$ denote the indicator function. Now let $\varphi(t) = (t-1)\mathbb{1}\{1 \leq t \leq 2\} + \mathbb{1}\{t > 2\}$. Clearly,

$$\mathbb{1}\{t \geq 2\} \leq \varphi(t) \leq \mathbb{1}\{t \geq 1\}. \tag{11}$$

We observe that,

$$\sup_{\Theta \in [-M,M]^{k \times p}} \sum_{\ell=1}^{L} \mathbb{1}\left\{(P - P_{B_\ell})f_\Theta > \epsilon\right\}$$

$$\leq \sup_{\Theta \in [-M,M]^{k \times p}} \sum_{\ell \in \mathcal{L}} \mathbb{1}\left\{(P - P_{B_\ell})f_\Theta > \epsilon\right\} + |\mathcal{O}|$$

$$\leq \sup_{\Theta \in [-M,M]^{k \times p}} \sum_{\ell \in \mathcal{L}} \varphi\left(\frac{2(P - P_{B_\ell})f_\Theta}{\epsilon}\right) + |\mathcal{O}|$$

$$\leq \sup_{\Theta \in [-M,M]^{k \times p}} \sum_{\ell \in \mathcal{L}} \mathbb{E}\varphi\left(\frac{2(P - P_{B_\ell})f_\Theta}{\epsilon}\right) + |\mathcal{O}|$$

$$+ \sup_{\Theta \in [-M,M]^{k \times p}} \sum_{\ell \in \mathcal{L}} \left[\varphi\left(\frac{2(P - P_{B_\ell})f_\Theta}{\epsilon}\right) - \mathbb{E}\varphi\left(\frac{2(P - P_{B_\ell})f_\Theta}{\epsilon}\right)\right]. \tag{12}$$

To bound $\sup_{\Theta \in [-M,M]^{k \times p}} \sum_{\ell=1}^{L} \mathbb{1}\left\{(P - P_{B_\ell})f_\Theta > \epsilon\right\}$, we will first bound the quantity $\mathbb{E}\varphi\left(\frac{2(P - P_{B_\ell})f_\Theta}{\epsilon}\right)$. We observe that,

$$\mathbb{E}\varphi\left(\frac{2(P - P_{B_\ell})f_\Theta}{\epsilon}\right) \leq \mathbb{E}\left[\mathbb{1}\left\{\frac{2(P - P_{B_\ell})f_\Theta}{\epsilon} > 1\right\}\right] = \mathbb{P}\left[(P - P_{B_\ell})f_\Theta > \frac{\epsilon}{2}\right]$$

$$\leq \exp\left\{-\frac{b\epsilon^2}{32\tau_{\boldsymbol{\alpha},k}^2 H_p^2 M^4 k^2 p^2}\right\} \tag{13}$$

We now turn to bounding the term

$$\sup_{\Theta \in [-M,M]^{k \times p}} \sum_{\ell \in \mathcal{L}} \left[\varphi\left(\frac{2(P - P_{B_\ell})f_\Theta}{\epsilon}\right) - \mathbb{E}\varphi\left(\frac{2(P - P_{B_\ell})f_\Theta}{\epsilon}\right)\right].$$

Appealing to Theorem 26.5 of [5] we observe that, with probability at least $1 - e^{-2L\delta^2}$, for all $\Theta \in [-M,M]^{k \times p}$,

$$\frac{1}{L}\sum_{\ell \in \mathcal{L}} \varphi\left(\frac{2(P - P_{B_\ell})f_\Theta}{\epsilon}\right)$$

$$\leq \mathbb{E}\left[\frac{1}{L}\sum_{\ell \in \mathcal{L}} \varphi\left(\frac{2(P - P_{B_\ell})f_\Theta}{\epsilon}\right)\right] + 2\mathbb{E}\left[\sup_{\Theta \in [-M,M]^{k \times p}} \frac{1}{L}\sum_{\ell \in \mathcal{L}} \sigma_\ell \varphi\left(\frac{2(P - P_{B_\ell})f_\Theta}{\epsilon}\right)\right] + \delta. \tag{14}$$

Here $\{\sigma_\ell\}_{\ell\in\mathcal{L}}$ are i.i.d. Rademacher random variables. Let $\{\xi_i\}_{i=1}^n$ be i.i.d. Rademacher random variables, independent form $\{\sigma_\ell\}_{\ell\in\mathcal{L}}$. From equation (14), we get,

$$\frac{1}{L}\sup_{\boldsymbol{\Theta}\in[-M,M]^{k\times p}}\sum_{\ell\in\mathcal{L}}\left[\varphi\left(\frac{2(P-P_{B_\ell})f_{\boldsymbol{\Theta}}}{\epsilon}\right)-\mathbb{E}\varphi\left(\frac{2(P-P_{B_\ell})f_{\boldsymbol{\Theta}}}{\epsilon}\right)\right]$$

$$\leq 2\mathbb{E}\left[\sup_{\boldsymbol{\Theta}\in[-M,M]^{k\times p}}\frac{1}{L}\sum_{\ell\in\mathcal{L}}\sigma_\ell\varphi\left(\frac{2(P-P_{B_\ell})f_{\boldsymbol{\Theta}}}{\epsilon}\right)\right]+\delta$$

$$\leq\frac{4}{L\epsilon}\mathbb{E}\left[\sup_{\boldsymbol{\Theta}\in[-M,M]^{k\times p}}\sum_{\ell\in\mathcal{L}}\sigma_\ell(P-P_{B_\ell})f_{\boldsymbol{\Theta}}\right]+\delta. \tag{15}$$

Equation (15) follows from the fact that $\varphi(\cdot)$ is 1-Lipschitz and appealing to Lemma 26.9 of [5]. We now consider a "ghost" sample $\mathcal{X}'=\{\boldsymbol{X}'_1,\ldots,\boldsymbol{X}'_n\}$, which are i.i.d. and follow the probability law $P$. Thus, equation (15) can be further shown to give

$$=\frac{4}{L\epsilon}\mathbb{E}\left[\sup_{\boldsymbol{\Theta}\in[-M,M]^{k\times p}}\sum_{\ell\in\mathcal{L}}\sigma_\ell\mathbb{E}_{\mathcal{X}'}\left((P'_{B_\ell}-P_{B_\ell})f_{\boldsymbol{\Theta}}\right)\right]+\delta$$

$$\leq\frac{4}{L\epsilon}\mathbb{E}\left[\sup_{\boldsymbol{\Theta}\in[-M,M]^{k\times p}}\sum_{\ell\in\mathcal{L}}\sigma_\ell(P'_{B_\ell}-P_{B_\ell})f_{\boldsymbol{\Theta}}\right]+\delta$$

$$=\frac{4}{L\epsilon}\mathbb{E}\left[\sup_{\boldsymbol{\Theta}\in[-M,M]^{k\times p}}\sum_{\ell\in\mathcal{L}}\sigma_\ell\frac{1}{b}\sum_{i\in B_\ell}(f_{\boldsymbol{\Theta}}(\boldsymbol{X}'_i)-f_{\boldsymbol{\Theta}}(\boldsymbol{X}_i))\right]+\delta$$

$$=\frac{4}{bL\epsilon}\mathbb{E}\left[\sup_{\boldsymbol{\Theta}\in[-M,M]^{k\times p}}\sum_{\ell\in\mathcal{L}}\sigma_\ell\sum_{i\in B_\ell}\xi_i(f_{\boldsymbol{\Theta}}(\boldsymbol{X}'_i)-f_{\boldsymbol{\Theta}}(\boldsymbol{X}_i))\right]+\delta \tag{16}$$

$$=\frac{4}{n\epsilon}\mathbb{E}\left[\sup_{\boldsymbol{\Theta}\in[-M,M]^{k\times p}}\sum_{\ell\in\mathcal{L}}\sum_{i\in B_\ell}\sigma_\ell\xi_i(f_{\boldsymbol{\Theta}}(\boldsymbol{X}'_i)-f_{\boldsymbol{\Theta}}(\boldsymbol{X}_i))\right]+\delta$$

$$\leq\frac{4}{n\epsilon}\mathbb{E}\left[\sup_{\boldsymbol{\Theta}\in[-M,M]^{k\times p}}\sum_{\ell\in\mathcal{L}}\sum_{i\in B_\ell}\sigma_\ell\xi_i(f_{\boldsymbol{\Theta}}(\boldsymbol{X}'_i)+f_{\boldsymbol{\Theta}}(\boldsymbol{X}_i))\right]+\delta$$

$$=\frac{4}{n\epsilon}\mathbb{E}\left[\sup_{\boldsymbol{\Theta}\in[-M,M]^{k\times p}}\sum_{i\in\mathcal{J}}\gamma_i(f_{\boldsymbol{\Theta}}(\boldsymbol{X}'_i)+f_{\boldsymbol{\Theta}}(\boldsymbol{X}_i))\right] \tag{17}$$

$$=\frac{8}{n\epsilon}\mathbb{E}\left[\sup_{\boldsymbol{\Theta}\in[-M,M]^{k\times p}}\sum_{i\in\mathcal{J}}\gamma_i f_{\boldsymbol{\Theta}}(\boldsymbol{X}_i)\right]+\delta$$

$$\leq\frac{8}{n\epsilon}48\sqrt{\pi}\tau_{\boldsymbol{\alpha},k}H_p M^2(kp)^{3/2}\sqrt{|\mathcal{J}|}+\delta \tag{18}$$

$$\leq\frac{384}{n\epsilon}\sqrt{\pi}\tau_{\boldsymbol{\alpha},k}H_p M^2(kp)^{3/2}\sqrt{|\mathcal{I}|}+\delta. \tag{19}$$

Equation (16) follows from observing that $(f_{\boldsymbol{\Theta}}(\boldsymbol{X}'_i)-f_{\boldsymbol{\Theta}}(\boldsymbol{X}_i))\overset{d}{=}\xi_i(f_{\boldsymbol{\Theta}}(\boldsymbol{X}'_i)-f_{\boldsymbol{\Theta}}(\boldsymbol{X}_i))$. In equation (17), $\{\gamma_i\}_{i\in\mathcal{J}}$ are independent Rademacher random variables due to their construction. Equation (18) follows from appealing to Theorem 3.1. Thus, combining equations (14), (15), and (19), we conclude that, with probability of at least $1-e^{-2L\delta^2}$,

$$\sup_{\boldsymbol{\Theta}\in[-M,M]^{k\times p}}\sum_{\ell=1}^L\mathbb{1}\left\{(P-P_{B_\ell})f_{\boldsymbol{\Theta}}>\epsilon\right\}$$

$$\leq L\left(\exp\left\{-\frac{b\epsilon^2}{32\tau_{\boldsymbol{\alpha},k}^2 H_p^2 M^4 k^2 p^2}\right\}+\frac{|\mathcal{O}|}{L}+\frac{384}{n\epsilon}\sqrt{\pi}\tau_{\boldsymbol{\alpha},k}H_p M^2(kp)^{3/2}\sqrt{|\mathcal{I}|}+\delta\right). \tag{20}$$

We choose $\delta = \frac{2}{4+\eta} - \frac{|\mathcal{O}|}{L}$ and

$$\epsilon = 2\max\left\{\sqrt{32\tau_{\boldsymbol{\alpha},k}^2 H_p^2 M^4 \log\left(\frac{4(\eta+4)}{\eta}\right)}\,kp\sqrt{\frac{L}{n}}, \frac{1536(\eta+4)\tau_{\boldsymbol{\alpha},k}H_p M^2\sqrt{\pi}}{\eta}(kp)^{3/2}\frac{\sqrt{|\mathcal{I}|}}{n}\right\}.$$

This makes the right hand side of (20) strictly smaller than $\frac{L}{2}$. Thus, we have shown that

$$\mathbb{P}\left(\sup_{\boldsymbol{\Theta}\in[-M,M]^{k\times p}}(Pf_{\boldsymbol{\Theta}} - \text{MoM}_L^n(f_{\boldsymbol{\Theta}})) > \epsilon\right) \le e^{-2L\delta^2}.$$

Similarly, we can show that,

$$\mathbb{P}\left(\sup_{\boldsymbol{\Theta}\in[-M,M]^{k\times p}}(\text{MoM}_L^n(f_{\boldsymbol{\Theta}}) - Pf_{\boldsymbol{\Theta}}) > \epsilon\right) \le e^{-2L\delta^2}.$$

Combining the above two inequalities, we get,

$$\mathbb{P}\left(\sup_{\boldsymbol{\Theta}\in[-M,M]^{k\times p}}|\text{MoM}_L^n(f_{\boldsymbol{\Theta}}) - Pf_{\boldsymbol{\Theta}}| > \epsilon\right) \le 2e^{-2L\delta^2}.$$

In other words, with at least probability $1 - 2e^{-2L\delta^2}$,

$$\sup_{\boldsymbol{\Theta}\in[-M,M]^{k\times p}}|\text{MoM}_L^n(f_{\boldsymbol{\Theta}}) - Pf_{\boldsymbol{\Theta}}|$$

$$\le 2\max\left\{\sqrt{32\tau_{\boldsymbol{\alpha},k}^2 H_p^2 M^4 \log\left(\frac{4(\eta+4)}{\eta}\right)}\,kp\sqrt{\frac{L}{n}}, \frac{1536(\eta+4)\tau_{\boldsymbol{\alpha},k}H_p M^2\sqrt{\pi}}{\eta}(kp)^{3/2}\frac{\sqrt{|\mathcal{I}|}}{n}\right\}$$

$$\lesssim \tau_{\boldsymbol{\alpha},k}H_p\max\left\{kp\sqrt{\frac{L}{n}}, (kp)^{3/2}\frac{\sqrt{|\mathcal{I}|}}{n}\right\}.$$

$\square$

## C.3 Proof of Corollary 3.5

*Proof.* We observe the follwoing.

$$|Pf_{\widehat{\boldsymbol{\Theta}}_n^{(\text{MoM})}} - Pf_{\boldsymbol{\Theta}^*}|$$

$$= Pf_{\widehat{\boldsymbol{\Theta}}_n^{(\text{MoM})}} - Pf_{\boldsymbol{\Theta}^*}$$

$$= Pf_{\widehat{\boldsymbol{\Theta}}_n^{(\text{MoM})}} - \text{MoM}_L^n(f_{\widehat{\boldsymbol{\Theta}}_n^{(\text{MoM})}}) + \text{MoM}_L^n(f_{\widehat{\boldsymbol{\Theta}}_n^{(\text{MoM})}}) - \text{MoM}_L^n(f_{\boldsymbol{\Theta}^*}) + \text{MoM}_L^n(f_{\boldsymbol{\Theta}^*}) - Pf_{\boldsymbol{\Theta}^*}$$

$$\le Pf_{\widehat{\boldsymbol{\Theta}}_n^{(\text{MoM})}} - \text{MoM}_L^n(f_{\widehat{\boldsymbol{\Theta}}_n^{(\text{MoM})}}) + \text{MoM}_L^n(f_{\boldsymbol{\Theta}^*}) - Pf_{\boldsymbol{\Theta}^*} \qquad (21)$$

$$\le 2\sup_{\boldsymbol{\Theta}\in[-M,M]^{k\times p}}|\text{MoM}_L^n(f_{\boldsymbol{\Theta}}) - Pf_{\boldsymbol{\Theta}}|$$

$$\lesssim \tau_{\boldsymbol{\alpha},k}H_p\max\left\{kp\sqrt{\frac{L}{n}}, (kp)^{3/2}\frac{\sqrt{|\mathcal{I}|}}{n}\right\}.$$

Inequality (21) follows from the fact that $\text{MoM}_L^n(f_{\widehat{\boldsymbol{\Theta}}_n^{(\text{MoM})}}) \le \text{MoM}_L^n(f_{\boldsymbol{\Theta}^*})$, by definition of $\widehat{\boldsymbol{\Theta}}_n^{(\text{MoM})}$.

$\square$

## C.4 Proof of Corollary 3.6

*Proof.* In this case, $H_p = 2$. Thus, the bound in Corollary 3.5 becomes $|Pf_{\widehat{\boldsymbol{\Theta}}_n^{(\text{MoM})}} - Pf_{\boldsymbol{\Theta}^*}| \lesssim \max\left\{\sqrt{\frac{L}{n}}, \frac{\sqrt{|\mathcal{I}|}}{n}\right\}$. By A7, $2e^{-2L\delta^2} = o(1)$. Thus,

$$\mathbb{P}\left(|Pf_{\widehat{\Theta}_n^{(\text{MoM})}} - Pf_{\Theta^*}| = O\left(\max\left\{\sqrt{\tfrac{L}{n}}, \tfrac{\sqrt{|\mathcal{I}|}}{n}\right\}\right)\right) \geq 1 - o(1). \text{ Hence, } |Pf_{\widehat{\Theta}_n^{(\text{MoM})}} - Pf_{\Theta^*}| = O_P\left(\max\left\{\sqrt{\tfrac{L}{n}}, \tfrac{1}{\sqrt{n}}\right\}\right)$$

Under A7, $\max\left\{\sqrt{\tfrac{L}{n}}, \tfrac{\sqrt{|\mathcal{I}|}}{n}\right\} \leq \max\left\{\sqrt{\tfrac{L}{n}}, \tfrac{1}{\sqrt{n}}\right\} = o(1) \implies |Pf_{\widehat{\Theta}_n^{(\text{MoM})}} - Pf_{\Theta^*}| = o_P(1)^2.$

Thus, $Pf_{\widehat{\Theta}_n^{(\text{MoM})}} \xrightarrow{P} Pf_{\Theta^*}$. Now, for any $\epsilon, \delta > 0$, $\mathbb{P}(Pf_{\widehat{\Theta}_n} \leq Pf_{\Theta^*} + \epsilon) \geq 1 - \delta$, if $n$ is large. From assumption A7, $\mathbb{P}(\text{diss}(\widehat{\Theta}_n, \Theta^*) \leq \eta) \geq 1 - \delta$ for any prefixed $\eta > 0$, and $n$ large. Thus, $\text{diss}(\widehat{\Theta}_n, \Theta^*) \xrightarrow{P} 0$, which proves the result. $\qquad\qquad\square$

## D  Additional Experiments

### D.1  Additional Simulations

**Experiment 3**  We use the same simulation setting as Experiment 1. However, the outliers are now generated from a Gaussian as well with mean coordinate $20 \times \mathbf{1}_5$, and covariance matrix $0.1I_5$, where $\mathbf{1}_5$ is the 5 dimensional vector of all 1's and $I_5$ is the $5 \times 5$ identity matrix.

**Experiment 4**  We use the same simulation setting as Experiment 2. However, the outliers are now generated from the same scheme as in Experiment 3.

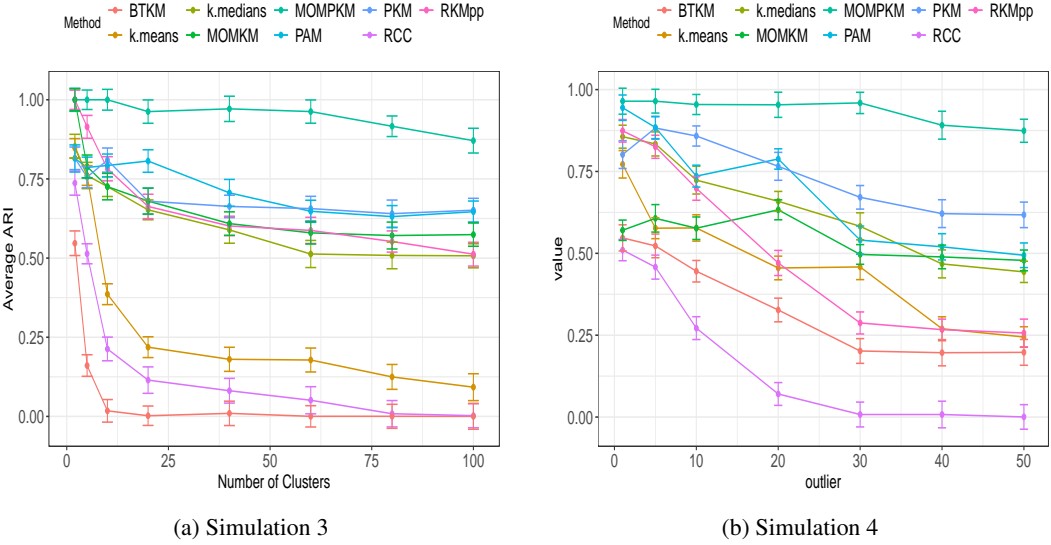

(a) Simulation 3                              (b) Simulation 4

Figure 1: Results on Simulation Studies based on Average ARI Values

### D.2  Case Study on Real Data: KDDCUP

In this section, we assess the performance of real data through the analysis of KDDCUP dataset [1], and consists of approximately 4.9M observations depicting connections of sequences of TCP packets. The features are normalized to have zero mean and unit standard deviation. The data contains 23 classes, out of which, the three largest contain $98\%$ of the observations. Following the footsteps of [2], the remaining 20 classes consisting of 8752 points are considered as outliers. We run all the algorithms as described in the beginning of section 4. The parameters considered for our algorithm are $L = 10000$, $\eta = 1.02$ and $\alpha = 1$. We measure the performance of this algorithm in terms of the ARI, as well as average precision and recall [3]. The last two indices are added following [2]. We report the average of these indices out of 20 replications in Table 1. For all these indices, a higher

---

$^2 X_n = o_P(a_n)$ if $X_n/a_n \xrightarrow{P} 0.$

Table 1: Results on KDDCUP Dataset

| Index | k.means | BTKM | RCC | PAM | RKMpp | PKM | MOMKM | MOMPKM |
|---|---|---|---|---|---|---|---|---|
| ARI | $10^{-5}$ | 0.01 | $10^{-5}$ | $10^{-16}$ | 0.81 | 0.24 | 0.76 | **0.87** |
| Precision | 0.25 | 0.23 | 0.19 | 0.23 | 0.64 | 0.43 | 0.56 | **0.71** |
| Recall | 0.00 | 0.14 | 0.07 | 0.11 | 0.63 | 0.49 | 0.59 | **0.76** |

value implies superior performance. Table 1 shows similar trends as discussed in Section 4 of the main text. In particular, MOMPKM resembles the ground truth compared to the state-of-the-art. Surprisingly, RKMpp performs better than other competitors (except for MOMPKM), which was not always the case for simulated data under ideal model assumptions. This is possibly because of the fact that the data contains only 47 features, compared to almost 5M samples, significantly capitalizing on the higher signal-to-noise-ratio, compared to that of the data used in the simulation studies.

## E    Machine Specifications

The simulation studies were undertaken on an HP laptop with Intel(R) Core(TM)i3-5010U 2.10 GHz processor, 4GB RAM, 64-bit Windows 8.1 operating system in R and python 3.7 programming languages. The real data experiments were undertaken on a cluster. The cluster has 656 cores (essentially CPUs) spread across a number of nodes of varying hardware specifications and ages. 256 of the cores are in the 'low' partition. There are 32 cores and 256 GB RAM per node.

## F    Ethics Statement

Our work focuses on algorithmic and theoretical contributions to unsupervised learning of data that feature outliers, unifying different center-based clustering frameworks. There are no immediate privacy or ethical concerns, but by addressing the persistent problem of presence of outliers, broader impacts extend beyond methodological contributions when the interpretation of pattern discoveries from the output of unsupervised learning methods have wider implications. Clustering has been used for countless applications, including community detection, drug discovery, and gene identification for cancers and other diseases. In such settings where the interpretations and decisions based on clustering solutions have significant scientific and societal bearing, it is critical that the outliers are not mistaken as original data while solving for optimal solutions or baseline truth.

That said, we have been careful not to overstate our claims. While theoretical and empirical evidence supports that we can significantly reduce the effect of outliers, users should not view our method as a panacea for the problem. Our algorithm provides only a partial remedy to a long-standing challenge faced by clustering methods, and we emphasize it may eliminate some but not all biases that may affect interpretations and decisions based on solutions output by unsupervised algorithms.