# OpenReview forum: "Uniform Concentration Bounds toward a Unified  Framework for Robust Clustering"
_NeurIPS.cc/2021/Conference — NeurIPS 2021 Spotlight_

### Official Review · Reviewer_TRVt · 2021-07-10

**Rating:** 7
**Confidence:** 3

**Summary:**

The authors consider the minimization of a Median-of-Means (MoM) objective, which is suitable to yield minimizers of a centroid-based clustering which is more robust to outliers. The type of the centroid-based clustering is defined over the employed measurement of distances, which can be any  Bregman divergence. The considered MoM objective is hence suitable to encompass various $k$-means formalizations.

The authors formulate an Adagrad method to minimize the MoM objective, and analyze properties of the global minimizer of the MoM objective with respect to robustness to outliers. They apply concentration inequalities to compare the MoM global minimizer to the global minimizer of the expected value of the generalized distance of points to their centroid. Since outliers do not influence the expected value, a small distance of the MoM minimizer to the one of the expected value indicates robustness to outliers.
The authors provide an experimental analysis of the power $k$-means objective, optimized by the proposed optimization scheme on synthetic data and one real-world dataset.

**Ethical Concerns:**

There are no ethical concerns.

**Limitations And Societal Impact:**

The limitations, for example the applicability of the theoretical result, could be discussed more in detail, There are justifications for the assumptions, but there is no discussion in which cases these assumptions might fail. In addition, the translation from the theoretical result to practical applications is not discussed. How does a guarantee on the global minimizer translate to the local minimizers found by the existing algorithms? There does probably not exist a proper answer at this state, but I think it should still be mentioned.

**Main Review:**

The authors provide a thorough analysis of the global MoM minimizer with respect to the robustness to outliers and their empirical analysis shows that the optimization scheme can actually yield more robust results than comparable clustering methods.
Yet, in spite of all that, I am still on the fence. One of the main reasons for this is the presentation. This is a very technical paper and it is on top of its technicality not easy to follow because of overloaded variables, undefined functions, and errors (cf. minor issues). Then, the related work is not explicitly discussed. In between the text, there are fragments that hint at the improvement of the derived result in comparison to a related result, but there is no overview of the SOTA knowledge of MoM minimizers in general and those for clustering in particular. In that respect, I find it hard to assess the gain of knowledge and the potential impact of the provided theoretical result.

In addition, the practical aspects (the optimization framework and its experimental analysis) and the theoretical results are not really coming together. The experiments do only consider the power $k$-means implementation of the optimization framework, although one of the emphasized strengths of the framework is its general applicability to various $k$-means variants. There is no discussion of the relation of the found local minimizers in comparison to the global minimizer, analyzed in the theoretical part. I am missing an experiment where the number of data points is increased such that the practical convergence can be checked in comparison to the theoretical guarantee. The theoretical results apply regardless of the optimization framework because they address the global minimizer of the objective.

In summary, I wonder if the theoretical results on their own are contribution enough to vote for acceptance. In that case, the related work should be discussed more thoroughly and the paper should maybe just be a theoretical one. If the theoretical results do however not yield enough impact to vote for acceptance, then the experimental analysis should be extended. In that case, some of the Lemmata should be put in the appendix and also other $k$-means variants should be implemented and compared to. The effect of parameters should be evaluated and more datasets should be used. Maybe, this paper would also be better served by being submitted to a journal, such that the 11-page supplement can be integrated into the main body of the paper.

### Minor Issues/Suggestions
* The name $P$ is used for the objective of $k$-means in the introduction and the probability distribution.
* The term _convergence rate_ for the clustering method is misleading, since I believe that most people associate therewith the convergence rate of the optimization method. In addition, that sentence in L70 is not really true, since the convergence rate does not exactly refer to the results of clustering methods but to the global minimizers of their objectives.
* The objectives in Examples (L88) miss an $1/n$ in front. Not that it would really matter but to be consistent with the objective (1).
* Algorithm 1:
   * Step 2: $g_j^(t)\leftarrow ..$ ?
   * Step 3: $\lVert g_j^(t')\rVert^2_2$ in the denominator, likewise in L113
* L159: what is $\Psi(\Theta,Q)$?
* L213: what is $O_P(..)$?
* it should be $\in \mathcal{O}(..)$ and so on, not $=\mathcal{O}(..)$
* Did you try using synthetic data with more than 5 dimensions? How does it affect the results?
* The formatting of the proof sketch of Thm 3.4 makes it impossible to read with all the inline math.

*Supplement*
* L8: $H_{s,p,M}$?
* Lemma A.2 is not necessary/ well known, I believe
* Lemma A.3: the $sup$ is missing in the equations and explanations are missing
* L24: $\partial J(P_C(\theta))=..$ and why is it a set? Why is there talk of a subgradient when $d_\phi$ is differentiable?
* L25: it should be $-d_\phi(P_C(\theta),\theta)= \langle \nabla \phi.. $

## Updates after authors response
The authors provided a very thoughtful response which shows in particular that the presentation will be overhauled, which was one of my main concerns. Due to the authors' response, I will change my borderline assessment to accept. I very much like the idea that the authors intend to publish an extended journal version. For the Neurips submission, there is the issue of space. From _my side_ I would not request that the authors add more experiments (although I would really like to see this one experiment to observe the convergence of the bound in practice, maybe in the journal version then). I would rather focus on the presentation of the theoretical work and the connection from the theoretical results to practice. The other reviewers may contradict me if they think I am wrong here :)





**Time Spent Reviewing:**

6

---

> ### Author Response · Authors · 2021-08-10
> **Response to Reviewer TRVt**
>
> We thank the reviewer for their constructive comments and thoughtful assessment of our work. Before responding point-by-point, we emphasize that the thorough theoretical analysis of our unifying framework is indeed our primary contribution, and we hope you will agree that the theoretical contributions are already novel and significant. Because not all theory results translate to actionable, useful algorithms, we further focus part of the paper to demonstrating its empirical success by identifying a clean and efficient algorithm for its implementation, and through designing careful (albeit not completely exhaustive) experiments.
>
> As you fairly point out, the paper is quite long and technical, and your review will help to improve streamlining the exposition and clarifying our contributions. Though we believe the new theory results stand on their own merits, we hesitate to take your first suggestion of revising the paper to be solely a theoretical one, as the practical aspects and implementation will likely be relevant to a sizeable portion of the readership. However, we will aim to more clearly discuss both aspects, incorporating reviewer suggestions and clarifying the writing within the page length. This will include discussing the limitations of the current empirical studies and questions that remain open in terms of the resulting optimization problem. A careful rewriting of dense passages, removing definitions such as A2 from the main text, and relegating details such as in the proof sketch of Thm 3.4 to the Supplement should make this feasible. Additional comments are below:
>
> 1. There is little work in the field of providing concentration bounds for general center-based clustering methods let alone their robust MoM version. One objective of this paper is to show that different center-based clustering methods can be brought under the same umbrella and can be robustified using a general-purpose scheme. We agree that more detailed explanations of the gaps in the theory of the SOTA are warranted. Following your suggestions, we will add such discussions with complete references in the final version.
>
> 2. Indeed, $k$-means and its variants are known to be non-convex. though techniques such as annealing [49] may be effective to circumvent this difficulty. A complete theoretical understanding from an optimization perspective for such methods is not yet fully developed and global results are notoriously difficult to obtain. We will add these remarks in the final version and emphasize non-convexity as another reviewer also suggests. We will also include a simulation study that more closely illustrates our bounds ``in action" as you suggest in the final version.
>
> 3. As you suggest, we will likely prepare an extended journal version of this analysis. Given the recent interest and developments in robust center-based clustering in the community, we feel that the current contents of the paper with further improvements with your constructive remarks will be an interesting contribution to  NeurIPS.  Following your suggestions echoed by other reviewers, we will include additional experiments as well as competing methods in the final version of the paper for a more thorough empirical component.
>
> Minor comments for the main text:
> 1. We will change the notation $P(\cdot)$ to $f_{k\text{-means}}(\cdot)$ for clarity.
>
> 2. Yes, "convergence rate'' here refers to the rate for the global sample minimizers to the global population minimizers, and you rightly point out that this may be confusing to those who typically interpret this with reference to the optimization routine. We will clarify this, and that the rates in line 70 are to the same effect: pertaining to the minimizers under the formulations we mention, rather than the particular solutions one instance may converge to.
>
> 3. We have now checked for consistency for the $\frac{1}{n}$ in objectives and fixed several typos, and thank you for noting these.
>
> 4. We apologize for the notational mismatch: $\Psi(\Theta,Q)$ should be replaced by $\int f_{\Theta} dQ$ and we have fixed this and other typos.
>
> 5. As already defined in line 212, we say that $X_n = O_P(a_n)$ (here {$X_n : n \in \mathbb{N}$} 's are random variables and {$ a_n :n \in \mathbb{N}$}'s are scalars) if the sequence {$X_n/a_n: n \in \mathbb{N}$} is tight, or bounded in probability. $O_P$ notations are quite standard in probability. Kindly refer to [2] of the main text and https://en.wikipedia.org/wiki/Big_O_in_probability_notation for more details.
>
>
> 6. Though it is standard to write $f(n) = O(g(n))$ if $\lim\sup_{n \to \infty} f(n)/g(n) < \infty$, following your suggestion, we will write $\in O(\cdot)$ instead of $= O(\cdot)$ in the final version.
>
> 7. We thank the reviewer for this suggestion. In the main text, we have considered the KDDCUP dataset which contains 47 features. Following your suggestions, we will extend experiment 1 of the main text in $\mathbb{R}^{30}$ with cluster centroids derived uniformly from the unit cube. Preliminary results show the same trends as demonstrated in experiment 1, with MOMPKM noticeably outperforming the other peer methods.
> 8. We agree and plan to replace the proof sketch with a brief overview, using the space for additional discussions and better exposition.
>
>
> Minor comments for the supplement:
>
> 1. We apologize for the notational mixup. $H_{s,p,M}$ should be replaced by $H_p$.
>
> 2. We had included Lemma A.2 for completeness, but indeed the result is well known and we are happy to omit it.
>
> 3. We thank you for pointing out this typo and will correct it and add discussion in the revision.
>
> 4.Though the subgradient is always defined, indeed in the case of Bregman divergences we have smoothness and may simply take the unique derivative. We will write it as you suggest to avoid unnecessary generality and possible confusion.
>
> 5. We thank you for pointing out the typo. It will be corrected in the final version.

---

### Official Review · Reviewer_2EUB · 2021-07-13

**Rating:** 7
**Confidence:** 3

**Summary:**

The paper looks at the Median-of-Means estimation framework for clustering and shows that it covers a range of k-means variants. They attempt to unify different frameworks of center-based clustering and introduce an algorithm that uses an adaptive gradient-based MoM algorithm. Under i.i.d. sampling of the data, they provide the convergence rates and the error rates obtained do not require any restrictions on the relation between the dimensions of the input points and the number of input points.

**Ethical Concerns:**

No major ethical concerns as far as I can see.

**Limitations And Societal Impact:**

They discuss in Section 5, the performance of MoM (in terms of convergence rate) being slower compared to other methods. Also, they discuss the dependency on the number of outliers (on how the number of outliers scale against the $n$ and state that if it is proportional to $n$ the error does not give a meaningful result. They also make a statement about the societal impacts in the checklist but I feel that any negative concerns that could arise are native to almost any robust framework.

**Main Review:**

The paper is well written and provides a well-structured overview of the results. As far as I can see, the statements are well-founded and the mathematics holds true. They show that under standard assumptions, the estimate of the centroids given by the MoM framework approaches the optimal set of centroids as the size of the data is scaled.

**Time Spent Reviewing:**

2

---

> ### Author Response · Authors · 2021-08-10
> **Response to Reviewer 2EUB**
>
> We thank the reviewer for the encouraging words, thoughtful remarks, and measured discussion of limitations. Indeed, we provide a careful theoretical analysis revealing slower convergence rates than standard counterparts, but as you mention this is a typical "price to pay" for robust versions. We hope the careful methods of analysis and unifying robust framework will be similarly received with excitement by the NeurIPS readership.

---

### Official Review · Reviewer_kBcm · 2021-07-13

**Rating:** 7
**Confidence:** 4

**Summary:**

This paper proposed to modify the k-means problem (a more general version) using the principle of median of means (MoM), which leads to a better concentration and robustness against outliers.

**Limitations And Societal Impact:**

No apparent limitations of the paper.

**Main Review:**

Overall, I find this paper interesting and novel. It is very easy to read and the theoretical results are reasonable.
I only have a few small comments.

Q1. [Non-convexity of the problem]
In general, the minimization of $\Theta$ is a non-convex problem (like the usual k-meams) so algorithm 1 may stuck at a local optimum.
It is worth acknowledge this in the paper and mention that we will re-initialize the starting points.


Q2. [Possibility to weaken compact support assumption?]
A feature  of MoM is that we may obtain exponential/Gaussian concentration even if the original random variables only have moment conditions (rather than sub-Gaussian).
So it allows a much heavier tail distribution.
I am wondering if the same thing is applicable here or not?

Under the k-means scenario (squares loss), it seems that we may relax the compact support assumption to be some moment conditions.
Any comments about this?


Q3. [Choice of L to be \log n]
Just curious: it seems that we can always choose $L \asymp \log n$, which will lead to the almost parametric rate (as indicated in the Remark of page 8).
Is there any danger of doing so?


**Time Spent Reviewing:**

6

---

> ### Author Response · Authors · 2021-08-10
> **Response to Reviewer kBcm**
>
> We thank the reviewer for the valuable comments and positive reception of our work. We address your questions as follows:
>
> Q1. As you rightly point out, since $k$-means and many variants entail non-convex problems, they are prone to stopping at local optima. We will highlight this point more clearly and explicitly mention random initializations, as currently it is only mentioned in passing after the empirical study. We will add discussion that we observed the robust version of power $k$-means to overcome this effect empirically to an extent, though establishing guarantees on reaching the global minimizer requires considerable efforts beyond the scope of the paper. This discussion will be included in the final version, following your suggestions.
>
> Q2. We thank the reviewer for the insightful comment. Indeed, it might be possible to extend the results for noise distributions that satisfy only moment conditions. Recent works in convex clustering (Tan and Witten, 2015) often use a sub-gaussian error model to obtain error rates. The work by Biau et al. (2008) and more recent work by Klochkov et al. (2020) obtain similar error rates under finite second-moment conditions. However, one has to artificially impose the conditions that the sample cluster centroids, $\widehat{\Theta}_n$ remain bounded. The bounded support assumption gives us a natural way to ensure that this is the case for a broader variety of Bregman divergences by appealing to the obtuse angle properties of Bregman divergences (Lemma 3.1 of the main text). Having said that, it might indeed be possible to extend our theoretical analysis to a more broader class of distributions may be appealing to results in local Rademacher complexities (Bartlett et al. 2005). We thank the reviewer for this stimulating discussion and will add it in the final version as a possible future direction in understanding center-based clustering in general.
>
> Q3. Unfortunately, it may not always be possible to choose $L=O(\log n)$. As mentioned in line 284 of the main text that if the model is contaminated only with $|\mathcal{O}|=O(\log n)$ many outliers, then one can choose $L$ in such a way that $L=O(\log n)$ and satisfying assumption A6. If $|\mathcal{O}|=\omega(\log n)$ (we say that $ f(n) = \omega (g(n))$ if $\lim\inf_{n \to \infty} f(n)/g(n) = \infty$), choosing $L=O(\log n)$ would possibly not negate the negative effect of the outlying observations and thus the rate of $\widetilde{O}(n^{-1/2})$ would not hold. We will add this discussion in the final version for clarity.
>
> References:
>
>  Tan, K.M. and Witten, D., 2015. Statistical properties of convex clustering. Electronic journal of statistics, 9(2), p.2324.
>
> Biau, G., Devroye, L. and Lugosi, G. (2008). On the performance of clustering in Hilbert spaces, IEEE Transactions on Information Theory 54(2): 781–790.
>
> Klochkov, Y., Kroshnin, A. and Zhivotovskiy, N., 2020. Robust $ k $-means Clustering for Distributions with Two Moments. arXiv preprint arXiv:2002.02339.
>
> Bartlett, P.L., Bousquet, O. and Mendelson, S., 2005. Local rademacher complexities. The Annals of Statistics, 33(4), pp.1497-1537.

---

### Official Review · Reviewer_w8MY · 2021-07-16

**Rating:** 7
**Confidence:** 4

**Summary:**

The paper provides a novel method for center-based clustering of i.i.d. points in $\mathbb{R}^d$ under a Bregman divergence. The key idea of the framework is to employ a median of means estimator defined over the objective function, i.e., the data set is partitioned into sets of equal size and the median of the objective values of the elements of the partition is minimized to obtain the clustering. The aim of the paper is to develop a clustering scheme that is more robust against outliers than common $k$-means and its variants/extensions and admits theoretical guarantees. To compute the clustering the well-known Adagrad gradient descent algorithm is used.

The main contribution of the paper is a rigorous theoretical analysis, proving consistency and convergence, i.e., the minimizer of the sample approaches the minimizer of the population as the sample size tends to infinity, at a bounded rate of order $(\sqrt{n})^{-1}$. The efficacy of the novel clustering method is experimentally evaluated and compared to other state of the art clustering algorithms.


**Limitations And Societal Impact:**

The authors do not explicitly discuss limitations of their work. I see no direct path to negative societal impact.

**Main Review:**

The problem is clearly well-motivated and the paper provides substantial progress in the direction of robust Bregman clustering. I find the experiments meaningful and agree that the novel method outperforms the methods it was compared to. The theoretical results are also compelling to me and underline the viability of the scheme.

I only have some minor questions/comments:
- In the proof of Lemma 3.1 you say that it can be observed that $ d_\phi(x, \theta) - d_\phi(x, P_\mathcal{C}(\theta)) + d_\phi(P_\mathcal{C}(\theta), \theta) \geq 0 $ implies the claim, which is $d_\phi(x, \theta) \geq  d_\phi(x, P_\mathcal{C}(\theta)) + d_\phi(P_\mathcal{C}(\theta), \theta)$. This is clearly not the case. Is this a typo or a major error (the remainder of the analysis depends on this lemma)? - I was out of time to check it myself.
- In the proof of Theorem 3.2, where does the $\sqrt{\frac{\log(2\delta)}{2n}}$ term come from? I assume it stems from some basic inequality like Markov's or some even more basic bound, but this is not immediate to me. Can you explain? Besides it must be $\log(2/\delta)$ (in some other places too).
- Since you are aiming at robustness it would be interesting to compare to a natural robust clustering scheme like $k$-median too.

**Time Spent Reviewing:**

5

---

> ### Author Response · Authors · 2021-08-10
> **Response to Reviewer w8MY**
>
> We thank the reviewer for the positive remarks and constructive comments. We respond individually to your questions below:
>
> 1.You are correct that in Lemma 3.1, it should read: $d_\phi(x,\theta) - d_\phi(x,P_{\mathcal{C}}(\theta)) - d_\phi(P_{\mathcal{C}}(\theta),\theta) \ge 0$; this is a purely typographical mistake and is now consistent in the context of the rest of the proof. This will be rectified in the revised version.
>
> 2. We again thank the reviewer for carefully pointing out the minor typo. Indeed these terms should be $\log(2/\delta)$. It comes from the bounds on the uniform deviation via Rademacher complexities (see for instance Theorem 2 of http://web.eecs.umich.edu/~cscott/past_courses/eecs598w14/notes/10_rademacher.pdf). These will be corrected in the final version.
>
> 3. To make sure that all the competitor algorithms are on equal footing, we compared our method to those clustering algorithms which use squared-error loss functions. But as pointed out by the reviewer, for completeness of the experiments we will include the popular $k$-medians method for a more complete comparison. Some initial results on simulations indicate a performance similar to $k$-means with no major improvements: for instance, in experiment 1 with $25\%$ outliers and $k=3$, it performs marginally better (ARI 0.758) than $k$-means (ARI 0.730), but not as well as other robust methods we have considered. We will add these results to the reported results across all experimental settings in the final version.

---

### Decision · Program_Chairs · 2021-09-27

**Decision:**

Accept (Spotlight)

**Comment:**

The authors develop a unified framework for robust clustering using the Median of Means approach, and provide an analysis of the consistency and convergence of the algorithms derived through this approach. The framework applies to a diverse set of clustering problems, and empirical comparisons show convincingly that the proposed method is indeed more robust than the baselines considered. Given the wide usage of clustering methods, this work will be of wide interest to the community.